# Marine Robots Mapping the Present and the Past: Unraveling the Secrets of the Deep

**Nadir Kapetanović** [1,*] , **Antonio Vasilijević** [1] **, Đula Nađ** [1] **, Krunoslav Zubčić** [2] **and Nikola Mišković** [1]

1   Laboratory for Underwater Systems and Technologies (LABUST), Faculty of Electrical Engineering and Computing, University of Zagreb, Unska 3, 10000 Zagreb, Croatia; antonio.vasilijevic@fer.hr (A.V.); dula.nad@fer.hr (Đ.N.); nikola.miskovic@fer.hr (N.M.)
2   Department of Underwater Archaeology, Croatian Conservation Institute, Cvijete Zuzorić 43, 10000 Zagreb, Croatia; kzubcic@hrz.hr
*   Correspondence: nadir.kapetanovic@fer.hr

**Abstract:** Underwater cultural heritage sites are subject to constant change, whether due to natural forces such as sediments, waves, currents or human intervention. Until a few decades ago, the documentation and research of these sites was mostly done manually by diving archaeologists. This paper presents the results of the integration of remote sensing technologies with autonomous marine vehicles in order to make the task of site documentation even faster, more accurate, more efficient and more precisely georeferenced. It includes the integration of multibeam sonar, side scan sonar and various cameras into autonomous surface and underwater vehicles, remotely operated vehicle and unmanned aerial vehicle. In total, case studies for nine underwater cultural heritage sites around the Mediterranean region are presented. Each case study contains a brief archaeological background of the site, the methodology of using autonomous marine vehicles and sensors for their documentation, and the results in the form of georeferenced side-scan sonar mosaics, bathymetric models or reconstructed photogrammetric models. It is important to mention that this was the first time that any of the selected sites were documented with sonar technologies or autonomous marine vehicles. The main objective of these surveys was to document and assess the current state of the sites and to establish a basis on which future monitoring operations could be built and compared. Beyond the mere documentation and physical preservation, examples of the use of these results for the digital preservation of the sites in augmented and virtual reality are presented.

**Keywords:** marine robotics; autonomous surface vehicle; autonomous underwater vehicle; remotely operated vehicle; multibeam sonar; side-scan sonar; bathymetry; photogrammetry

## 1. Introduction

Knowledge about the Earth and its evolving environment is becoming increasingly important. However, we learn more about the stable surfaces of other planets than about our own planet. Rovers that have been traveling around Mars for years, landers on the remote Titan, and even the international missions to the Moon cannot hide the fact that the environment of the ocean floor is still completely mysterious, yet is only a few kilometers from our coasts. The oceans and seas are difficult to reach for direct observation. Only in the last 20 to 30 years have we succeeded in exploring and mapping the Earth's seabed, mainly through technological advances such as acoustic remote sensing.

Sometimes, new technologies take time to reach their full potential. Hence, technology is often a bridge between different disciplines. This paper presents a case study of nine UCHs that bridges the gap between the methods used exclusively in maritime archaeology, marine robotics and remote

sensing. The challenges in these areas are described here along with their possible solutions. Methods for recording and documentation of underwater cultural heritage sites (UCH) have developed considerably over the last two decades. The combined use of optical and acoustic technologies makes it possible to provide a high-quality digital 3D reconstruction of large and complex underwater scenarios [1,2]. These technologies create the possibility to study the UCH in onshore laboratories in a non-intrusive way [3]. The resulting digital reconstructions are often accepted for archaeological purposes and especially for documentation and monitoring activities [4,5]. These tools can be used to predict how sites have changed both recently and far in the past, but also how they might change in the future. These tangible results can be used as effective tools for public engagement.

Nothing has made as much progress in underwater archaeology as the equipment used for site and environment imaging. Recently, photogrammetry, photomodeling, simultaneous localisation and navigation (SLAM), organized light processing, multibeam and numerous other acoustic remote sensing techniques have been applied to underwater sites in the Mediterranean Sea [1,6–8]. However, even as archeologists strive to replace the laborious manual documentation process with more effective processes, no particular system has shown enough simple advantages to be generally accepted or recognized as the current standard for digital site reporting. Price, precision, durability and time issues in post-processing are usually of paramount importance. A common challenge for archeologists, who usually lack the expertise to process the data themselves, is the opportunity to incorporate multibeam sonar point clouds and photomosaics to produce publication-quality archaeologically relevant charts and maps.

Marine robotics is developing into a powerful tool for remote sensing in shallow seas and offers a wide range of possibilities for surveying without site disturbance (2.5D modelling of a site or landscape without excavation) [9]. Marine robots are generally not faced with the technical difficulties of underwater activities in these near-shore archaeological underwater environments, but potentially encounter a much greater challenge when they must cooperate with human divers [10]. Archaeologist scuba divers incorporate high versatility, intelligent coordination and a wide range of manual skills to reduce operating costs. However, these human advantages tend to disappear as the area to be explored becomes larger and deeper or the time for field operations becomes shorter.

Furthermore, robots can endure longer surveying missions without the fatigue that humans tend to experience, they can move faster and deeper, they do not risk human injury or fatalities, and their deployment is more cost-effective in the long run. They can also operate very well in cold and poorly visible/turbid environments and usually have a range of localisation and surveying sensors with a much greater range than that of a human and/or digital single lens reflex camera (DSLR).

A few decades ago, precise underwater localization was a great challenge and a big stepping stone to the use of autonomous underwater robots for detailed, precise surveying missions. However, the development of navigation systems for autonomous marine vehicles based on a combination of measurements from inertial measurement unit (IMU), Global Positioning System (GPS) and Doppler Velocity Logger (DVL) has greatly improved the precision of underwater localization [11]. In addition, underwater vehicles with integrated acoustic localization aiding systems like ultra-short baseline (USBL) offer a way for underwater exploration to go much deeper and longer than divers [12].

Underwater robot localization equipment and algorithms have been significantly improved, which means that the experts using the robots can acquire acoustic/visual data with precise placemarks and feed it directly into commercially available bathymetry and/or photogrammetry software. All of the technological improvements mentioned above have led to a significant increase in the use of robotic systems for surveying underwater sites, either with or instead of divers. These robotic systems include remotely operated vehicles (ROVs), autonomous surface vehicles (ASVs), autonomous underwater vehicles (AUVs) and unmanned aerial vehicles (UAVs) for inspections in shallow water. The applications of these vehicles range from underwater/maritime archaeology, biology, geology, security and many others.

This article contains case studies of the authors' recent activities regarding the use of autonomous marine vehicles for the survey of underwater archaeological sites. It presents a total of nine case studies on the use of marine robotics for survey applications at UCH sites in the Mediterranean. The methodology used for data collection is described for each site, as each of them had its own specificities. Depending on the site, the authors used all or some of the autonomous marine vehicles (ASV, AUV, ROV, UAV) for data collection. The results of the post-processed collected data are presented in the form of bathymetric maps/2.5D models and/or side-scan images and/or visual images taken by the AUV and/or ROV and/or UAV, as well as photogrammetry models and/or orthogonal photomosaics generated from the visual data. It is important to emphasize that this was the first time any of the underwater archaeological sites presented in this paper were recorded using modern UCH documentation methods, including visual and acoustic remote sensing techniques, as well as autonomous marine vehicles.

The long-term research objective of the authors is to develop artificial intelligence-based fully autonomous survey mission planning algorithms for survey applications in maritime archaeology. These missions can be performed by various marine vehicles, e.g., ASV, AUV, or ROV tethered to an ASV. Research objective for this paper was to examine best practices of autonomous systems and various sensors use for high-resolution georeferenced UCH site documentation and future preservation. This paper presents the results of authors' work in technical and research areas, including:

1.  the design and development of a small size robust ASV for bathymetric surveys;
2.  design of UCH site visual and acoustic documentation methodology with an ASV, AUV, ROV and UAV. The integration of remote sensing technologies with autonomous marine vehicles made the task of site documentation even faster, more accurate, more efficient and precisely georeferenced. It includes the integration of multibeam sonar, side scan sonar and various cameras into autonomous surface and underwater vehicles, remotely operated vehicle and unmanned aerial vehicle. In total, case studies for nine UCH sites around the Mediterranean region are presented. Each case study contains a brief archaeological background of the site, the methodology of using autonomous marine vehicles and sensors for their documentation, and the results in the form of georeferenced side-scan sonar mosaics, bathymetric models or reconstructed photogrammetric models.
3.  a discussion of the potential of using intelligent buoys/ASVs for the purpose of underwater diver localization and augmented reality UCH site guides, and
4.  a discussion of the UCH site documentation and preservation transgression from physical (in situ) to the virtual reality realm.

The rest of this article is structured as follows: A general overview of the authors' activities so far, including research, technical projects and the publications contained therein, is presented in Section 2. The autonomous marine vehicles and their sensors used for on-site underwater site recordings are presented in Section 3. Section 4 describes the processing pipeline for various types of data recorded at the chosen UCH sites. Remote sensing results of nine UCH sites surveys are presented in Section 5. The Section 6 brings new directions in the documentation, promotion and preservation of archeological underwater sites by transferring them into the realm of virtual reality. Concluding remarks are given in Section 7.

## 2. Prior Work

The Laboratory for Underwater Systems and Technologies (LABUST) has participated in many research and technical projects that have pushed the boundaries of the state-of-the-art in marine robotics and remote sensing applications in numerous scientific fields. For example, in the projects TRITON [13], ADRIAS [14] and BLUEMED [4,5], they have used marine robotics as a remote sensing tool for maritime archaeology; marine ecology in the projects subCULTron [15] and e-UReady4OS [12,16]; Maritime safety in the MORUS project [17]; Maritime inspection in the HEKTOR project; Promotion of diver safety and augmenting their performance in the projects CADDY [10] and ADRIATIC [18].

An overview of the geographical distribution of the sites where LABUST's marine robots were used for remote sensing is shown in Figure 1.

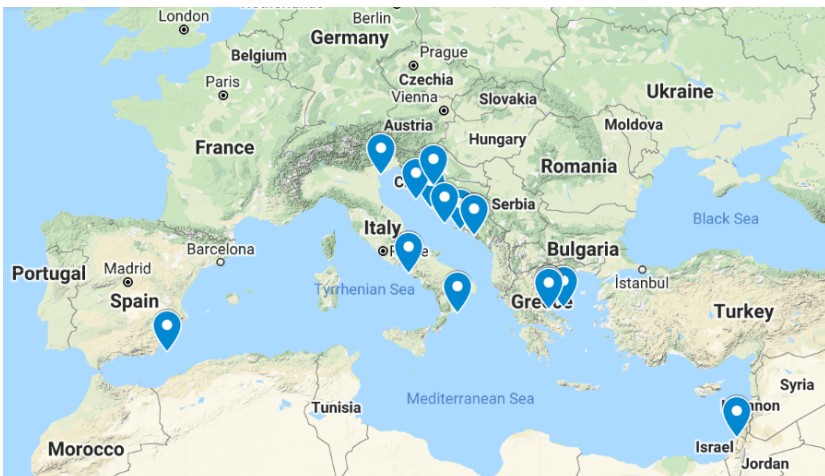

**Figure 1.** Google Map of locations at which Laboratory for Underwater Systems and Technologies (LABUST) used marine robots in remote sensing missions.

The long-term research goal of the authors is to develop the current unmanned survey/inspection missions by marine vehicles into missions that are performed by the marine vehicles in a fully autonomous manner controlled by artificial intelligence. This of course means that online sensor data processing must be developed to enable the vehicle to perceive its environment (as published in [19–21]), as well as mission and path planning algorithms, so that the behaviour of the vehicle is responsive to the new information about its environment, as published in [15,22–25]. For successful, fully autonomous reconnaissance missions, it is of utmost importance that the marine vessels estimate their position accurately, as published by the authors in [12,26,27].

## 3. Equipment

An autonomous surface vehicle (ASV) mounted with a multibeam sonar, which has been used for remote sensing survey missions, is described in more detail in Section 3.1. Specifications of the side-scan sonar-mounted autonomous underwater vehicles (AUV) are given in Section 3.2. The remotely operated vehicle (ROV) that was used for remote visual inspection of the sites at the seafloor is described in Section 3.3. The unmanned aerial vehicle (UAV) used for shore inspection close to the survey sites is briefly described in Section 3.4.

### 3.1. Autonomous Surface Vehicle

An ASV equipped with a Norbit WBMS 400/700KHz multibeam echosounder/sonar (MBES) and accompanying Applanix navigation system together with a high-precision Trimble GPS antennae was used to collect the acoustic data. This is one of the many application dependent versions of the so-called dynamic positioning platforms (PlaDyPos or H2Omni-X), see Figure 2a. The surface vehicle has been developed by LABUST and is used for a variety of applications, from support to underwater archaeology [5], as a dive monitoring platform that allows divers to navigate and monitor from the surface [28], as communication router between underwater and aerial vehicles [29], used in ASV swarms for long-term monitoring of the underwater environment [15], for mapping (obtaining photomosaic and bathymetry) of shallow water areas [2] and for mine countermeasures [30].

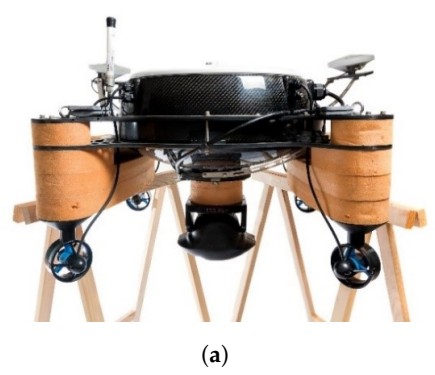

(**a**)

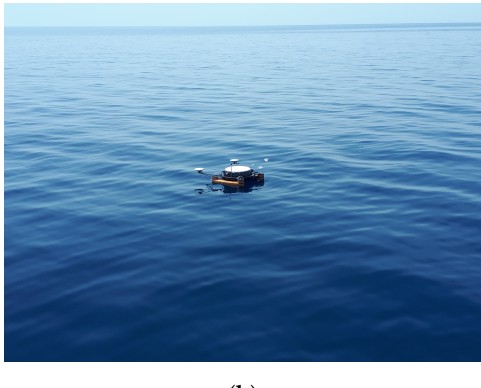

(**b**)

**Figure 2.** (**a**) Autonomous surface vehicle (ASV) PlaDyBath with sonar mounted below, Trimble Global Positioning System (GPS) antennae in the back and a WiFi antenna on the left. (**b**) ASV PlaDyBath during a monitoring mission.

The ASV is fully actuated with four electric thrusters that make up the X configuration. This configuration allows horizontal movement in any orientation. The ASV has a diagonal length of 1 m, is 0.35 m high and weighs about 20–30 kg depending on the payload configuration in the experiments. The maximum speed under ideal conditions is 1 m/s. Such a configuration of the vessel is very well suited for research purposes due to its simple deployment procedure, robustness under real environmental conditions and low energy consumption [28,31].

For acoustic remote sensing applications, however, the ASV was converted into a catamaran shape for better hydrodynamic performance, as shown in Figure 2a. This version of the surface platform is called Platform for Dynamic Bathymetric Imaging (PlaDyBath or H2Omni-H), as shown in Figure 2a. While ASV PlaDyBath performs its current survey mission autonomously for one part of the UCH, the operator is able to process MBES data collected during the previously surveyed UCH site part. These data are processed in low resolution using Qimera software to monitor data quality and coverage. As soon as the batteries are exhausted, the vehicle is lifted onto the working boat, the batteries are changed and high-quality sonar recordings are transferred to the ASV operator's computer. The ASV PlaDyBath was used for the surveying and 3D modeling of underwater cultural heritage sites (UCH) around the Mediterranean Sea, as shown in Figure 2b.

Mission planning and control were carried out with the open-source software Neptus, developed by the Laboratório de Sistemas e Tecnologia Subaquática (LSTS) of the University of Porto, Portugal. It is a distributed command and control infrastructure for the operation of all types of unmanned vehicles. Neptus supports the different phases of a typical mission life cycle: planning, simulation, execution and post-mission analysis. Neptus can be adapted by operators to mission-specific requirements and extended by developers through a comprehensive plug-in framework. Neptus communicates with the vehicle's hardware via so-called IMC messages, also developed by LSTS. In the case of PlaDyBath ASV, these mission-specific IMC messages are bypassed by a robot operating system (ROS) on-board the high-level computer, which in turn communicates with middleware microcontrollers to control the vessel's motors. The sonar is switched on both physically and via the network interface by the operator, who is connected to the vehicle via WiFi. The WBMS software used for sonar data acquisition on board the vessel was also controlled via a remote desktop over WiFi.

Multibeam Sonar

The Norbit iWBMSc multibeam sonar is the main sensor for bathymetric data acquisition, as shown in Figure 2a. The sonar is integrated with the latest GNSS-assisted inertial navigation system (Applanix SurfMaster), has 80 kH bandwidth, roll stabilization, an Ethernet interface and an integrated sound velocity measurement unit. The basic sonar features are 5–210° swath, adjustable swath tilt angle, 256–512 beams, 200–700 kHz frequency range, nominal frequency at 400 kHz, range 0.2–275 m

(160 m typical at 400 kHz). The ping rate goes up to 60 Hz or can be set to adaptive. The resolution longitudinal × transverse standard is $0.9 \times 1.9°$ at 400 kHz and $0.5 \times 1.0°$ at 700 kHz. The output data of this sonar are range and bearing measurements as well as backscatter data. In parallel, the Applanix module logs the navigational information (position and orientation in 3D). In the postprocessing phase, range and bearing measurements are merged with navigation data to generate the bathymetric point, which is then interpolated into a 2.5D surface.

### 3.2. Autonomous Underwater Vehicle

An autonomous underwater vehicle (AUV) is a type of mobile robot that, as the name suggests, is propelled through the water. The difference between an AUV and a ROV, both of which are controllable in all three dimensions, is that an AUV is controlled completely autonomously by its onboard computer. Lightweight autonomous underwater vehicle (LAUV) Lupis (see Figure 3a) has been purchased by LABUST from OceanScan—Marine Systems and Technology, Lda. The LAUV system was originally developed by the Laboratório de Sistemas e Tecnologia Subaquática (LSTS) from Porto University and was developed in cooperation with OceanScan—Marine Systems & Technology, Lda. The complete LAUV system includes all the equipment necessary for communication with the vehicle, the guidance and control software, external navigation aids and a set of optional devices to facilitate various operations. LAUV Lupis is a lightweight, modular platform prepared for the integration of a number of different sensors and sonars. The vehicle is designed for cost-effective oceanographic, environmental and inspection surveys to meet a wide range of applications. On the software side, LAUV Lupis uses the same LSTS toolchain (DUNE, IMC, and Neptus), as does the ASV PlaDyBath.

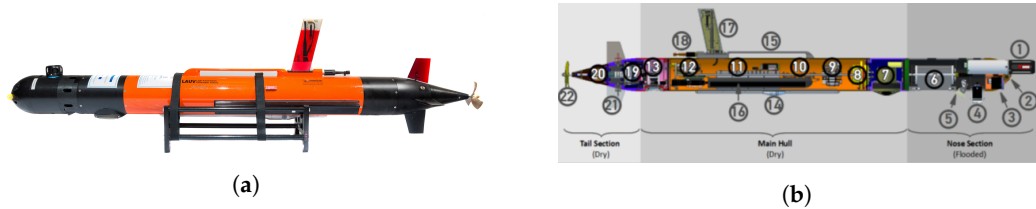

(a)                    (b)

**Figure 3.** (**a**) Lightweight autonomous underwater vehicle (LAUV) Lupis. (**b**) Blueprint of LAUV Lupis' sections and components.

LAUV Lupis is equipped with (see Figure 3b): (1) Environmental sensors (sound speed, fluorescence, temperature, conductivity, pressure turbidity...), (2) Emergency pinger, (3) Forward looking sonar (FLS), (4) Acoustic transducers (USBL and Acoustic modem), (5) Illumination module, (6) Multi-beam echo-sounder, (7) DVL, (8) IMU, (9) Acoustic modem (electronics), (10) On-board CPU & solid-state hard disk, (11) Batteries, (12) Communication and Navigation boards (Wi-Fi, GPS, Global System for Mobile Communications (GSM), Iridium, Compass), and (13) Camera, (14) Ballast weights, (15) Handle, (16) Side-scan sonar (SSS) transducers, (17) Flexible antennas mast (Wi-Fi, GPS, GSM, Iridium), and (18) Batteries charging connector, (19) Thruster and controllers, (20) Magnetic coupling, (21) Flexible fins, (22) Propeller.

Side-Scan Sonar

Instead of measuring the depth to the sea floor, as multibeam sonar does, the side-scan sonar provides details about the composition and structure of the sea floor. This is based on the intensity analysis of the echoes reflected from the seafloor. Normally, harder materials have a higher reflectivity than softer materials.

Klein's UUV-3500 side-scan sonar ,which is mounted on LAUV Lupis, is a high resolution dual frequency side-scan sonar that uses wideband signal processing techniques to achieve high-resolution imagery, even at long range. Low power consumption is achieved by using long frequency modulated chirp transmissions, while the integrated hardware signal processing generates acoustic

images with high fidelity. The sonar consists of sonar electronics and a pair of sonar transducers. The sonar electronics are packaged as an integrated part of an AUV electronics assembly. The sonar transducers are mounted on the outside of the AUV and configured with penetrators through the hull. The specifications of the side-scan sonar are given in Table 1. Output sonar data are loaded from AUV Lupis logs into Neptus Mission Review and Analysis software. It is possible to review the low and high frequency sonar logs in a waterfall fashion. Neptus applies time-varying gain (TVG) to correct range-intensity dependence. It also merges navigation estimation data (corrected with GPS fixes after the surfacing) with raw sonar data to produce georeferenced sonar imagery.

**Table 1.** Side-scan Sonar Specifications.

| Parameter | Value |
| --- | --- |
| *Center Frequencies (fc):* | 400 KHz LF and 775 KHz HF |
| *Transmit Pulse:* | Wideband Chirp; 1, 2, 4, 8 msec |
| *Sonar Range:* | 15, 20, 30, 40, 50, 60, 75, 100, 125, 150, 175, 200 m |
| *Across Track Resolution:* | varies with pulse length—1.2 cm, 2.4 cm, 4.8 cm, 9.6 cm, 19.2 cm |
| *Horizontal Beamwidth @fc:* | 0.34 degrees (LF and HF) |
| *Maximum Range:* | 100 @ HF; 200 m @ LF (conditions permitting) |
| *Depth Limit:* | 300 m |
| *Local Data Storage Optional:* | Dual Frequency: Max is 6.7 GB/h; Min is 0.4 GB/h. Single Frequency: Max is 3.4 GB/h; Min is 0.2 GB/h |
| *Operating Temperature:* | −2–50 °C [28–122 °F] |

*3.3. Remotely Operated Vehicle*

BlueROV2 (shown in Figure 4a) is a very powerful, flexible and affordable small-size ROV. The 6-thruster vector configuration, coupled with strong static stability, ensures a vehicle that is smooth and stable, yet easy to handle. BlueROV2 offers top-class mini ROV capabilities at the cost of the simplest commercial ROVs.

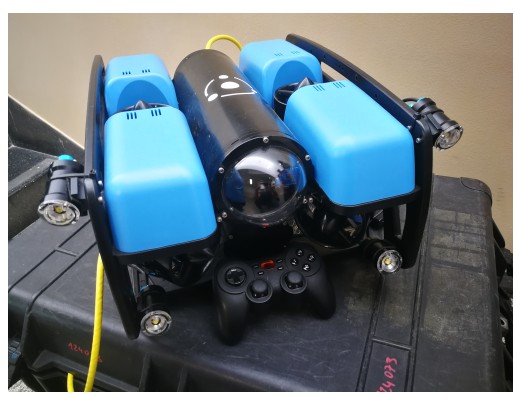
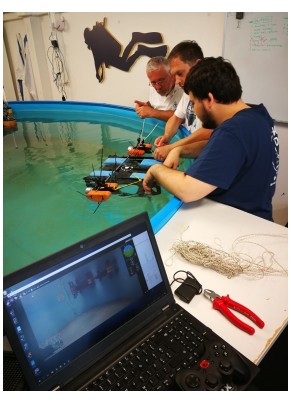

(**a**)                    (**b**)

**Figure 4.** (**a**) BLUEROV2 and its wireless controller. Courtesy of BlueRobotics. (**b**) Adjusting the neutral buoyancy and balancing the additional underwater lamps.

Its main features are: live 1080p HD video, highly controllable thrust vector configuration, stable and optimized for inspection class and research class missions, simple user interface, highly expandable with three free cable penetrations and additional mounting points, standard 100 m depth and up to 300 m tether, six T200 thrusters and basic electronic speed control (ESC) for a high thrust-to-weight ratio, quick-change batteries for all-day use, very flexible top case for a clean connection of BlueROV2 to a computer.

The vehicle is delivered in the form of a set and is partially assembled to make its use as easy and comfortable as possible. It consists of many other NIDO products, including T200 thrusters, 3.4 and 4 series waterproof enclosures, Bar30 pressure sensors, fathom-X tether interface (FXTI) and

fathom-tether interface electronics. Many of these products have been on the market for a long time and have proven themselves in thousands of successful field hours.

### 3.4. Unmanned Aerial Vehicle

The unmanned aerial vehicle (UAV) DJI Phantom 4 (see Figure 5) has two collision avoidance sensors at the front and rear of the drone and others below and at the edges. The camera starts recording at launch so that the drone can find a homing location when it is ready to return home. The camera hangs below the drone on a gimbal device that keeps it stable in all conditions in which it flies. The arms also have lights so that the drone can be quickly detected in the air.

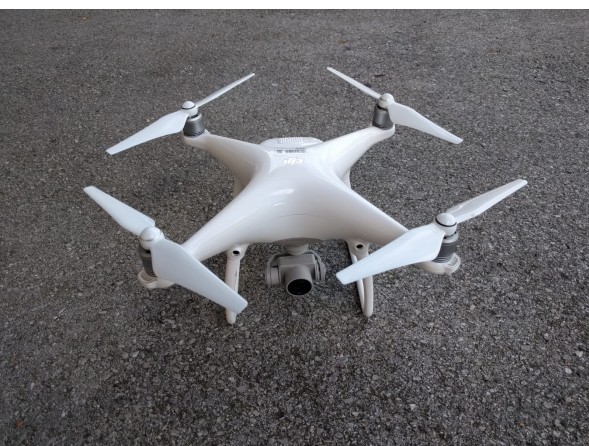

**Figure 5.** UAV DJI Phantom 4.

Inside the Phantom 4 shield is the high capacity battery with 5.87 Ah. This allows it to fly for up to half an hour. The Phantom 4 has a highly efficient system to avoid obstacles. In addition, sensors are mounted on the back and sides to detect possible dangers. The Phantom 4 has a remote controller with a built-in screen with a 5.5-inch monitor and a resolution of 1080p. The remote control connects to the drone at a distance of up to 7 km and can transmit the full HD video to the screen. The Phantom 4 features a 20-megapixel sensor with 12 levels of dynamic range, allowing for more precise shooting, even in extreme light. The automatic shutter prevents camera blur and makes shooting easier.

## 4. Data Acquisition and Processing Pipeline

### 4.1. Multibeam Sonar Data

When surveying with MBES, one should be aware of the physical limitations of sonar beam resolution, as well as the performance limit of spatial resolution of samples seafloor patches. The size of the "footprint" $\delta_b \times \delta_b$ which MBES beam ensonifies for beam width angle $\alpha$ at depth $h$ can be estimated as:

$$\delta_b = 2h \tan \frac{\alpha}{2}. \tag{1}$$

If we take that the mean across and alongtrack beam with Norbit WBMSc sonar is 1°, we obtain $\delta_b = 0.02h$. Furthermore, spatial resolution of the MBES $\delta_s$, i.e., the distance between adjacent sonar beams at the assumed flat seafloor, can be estimated as:

$$\delta_s = 2h \tan \frac{\psi}{2N} \tag{2}$$

for a given swath angle $\psi$, number of beams $N$ and depth $h$. For a swath angle $\psi = 90°$ and $N = 256$, which were used in most of the surveys, we obtain $\delta_s = 0.006h$. We used these equations to determine the upper bound of the bathymetric model interpolation grid cell size.

The autonomous surface vehicle (ASV) PlaDyBath's survey path designed lawnmower-shaped survey missions along and across the area of interest. In order to have as much quality coverage as possible, it is needed to overlap the adjacent lawnmower lanes with percentage $k \in (0, 1]$. The distance between the lawnmower lanes $d$ for a given overlap percentage can be estimated for the assumed approximately flat seafloor as:

$$d = 2 \left( 1 - \frac{k}{2} \right) h \tan \frac{\psi}{2}. \tag{3}$$

During the acquisition phase, there were issues with swath width angle of $\psi = 120°$, especially at 700 kHz frequency. This was due to the fact that the floater foams on port and starboard of the ASV blocked some of sonar beams or the sonar was not submerged deep enough below the ASV. This was solved by using the 400 kHz frequency for higher range, but with reduced swath angle at $\psi = 90°$. With swath angle of 90° and overlap $k = 0.5$, the distance between the lawnmower lanes was easily computed as $d = 1.5h$.

Norbit's WBMS software was used for logging raw MBES data, both for the point cloud as well as backscatter data. QPS Qimera was used for processing the raw sonar data. In order to process the data in Qimera, the .son format of raw sonar data decoupled from the navigation data had to be exported to .s7k format which was then loaded into Qimera. Applanix INS logged the navigation data into .pos format files that could be directly imported into Qimera. Based on the Applanix datasheet, the orientation data uncertainty was set to 0.3° for heading and 0.08° for roll and pitch. The heading and roll/pitch precisions were based on patch tests performed by the instructions given by Norbit. The precision of GPS position data was on the order of 10 cm when only Trimble antennae were used connected to the Applanix INS. This precision was improved to the order of 1 cm if NTrip client was connected through a 4G modem, when we obtained access to the local base station system. Both GPS and INS data were used to merge sonar data with navigation data and generate the bathymetric model. The bathymetric models were georeferences in UTM system.

Raw sonar pings had to be filtered for outliers in Qimera. This was done by limiting the range of depth values appropriate for the location as was read from the WBMS software. Swath Editor was used to select accrostrack ourliers for several tens of pings at the same time, while Slice Editor was used to filter our the pings on the outside of the delimited UCH area, as well as ping which was recorded while the ASV was turning from one lawnmower lane to the next. The turning motion causes the so-called fan-out effect, where the inside of the turning curve is sampled much more that the sparsely sample outside part of the turning path. This unevenly sampled data were not desired so it was deemed as outliers.

Interpolation was done using the default interpolant from Qimera. NOAA CUBE interpolation was also used but it did not show better results. The grid cell size for the interpolation was based on calculations of $\delta_s$, which was the lower limit of this parameter. Since Applanix INS does not have a heave sensor integrated, waves from the surface were translated directly to the bathymetric model. This was solved using the Wobble Analysis tool in Qimera as well as additional spline interpolation for larger areas with less details.

One of the objectives of the BLUEMED project was to generate an optoacoustic 3D model, which would merge high-resolution textured photogrammetric models with lower-resolution georeferenced bathymetric models. To this end, at every BLUEMED UCH site, a team of divers from University of Calabria would put a couple of acoustic markers that needed to be detected in the backscatter data of the MBES. This was needed to enable easy collocation of the photogrammetric and the bathymetric model. Unfortunately, for reasons unknown to us, the acoustic markers could not be detected in the backscatter data of the MBES so feature-based collocation algorithms had to be used in order to align these models.

*4.2. Side-Scan Sonar Data*

Survey missions for the LAUV Lupis mounted with the side-scan sonar were planned in a form of the lawnmower pattern at constant altitude from the seafloor. Distance between adjacent lawnmower lanes was determined with the assuming that the seafloor is locally approximately flat, and that the maximum slant range is ten times larger than the operating altitude. Since all UCH sites except for one were at depth of 6–30 m operating altitude was 2–3 m for safety reasons, so that the AUV does not collide with the artefacts at the seafloor if the amphorae/anchor mounds rose too suddenly from the flat surrounding. Assuming locally flat seafloor, the lawnmower lane width was set in the range 15–30 m. In some cases, we obtained significant surface returns because the AUV was relatively close to the surface, especially at Baiae UCH site where the depth was 8 m at most. The side-scan sonar was pinging at both low and high frequency so that we could detect the differences in the resolution and object detection methods. For the analysis, we of course used only high frequency data.

We do not have a direct access to the raw side-scan sonar data. Neptus Mission Review and Analysis software downloads the mission logs from the LAUV Lupis either over WiFi or from the USB located in the nose part of the vehicle. It then processes raw side-scan sonar data, merges it with the navigation estimation data, and presents it in the waterfall view. The position data represent the filtered dead reckoning estimation augmented with the DVL and GPS measurements when available. The path of the vehicle is corrected in the post-processing phase from the last stable GPS fix after resurfacing to the beginning of the dive. There is also an option for adjusting the time varying gain (TVG) as needed. Except for TVG, the brightness of side-scan sonar imagery can be equalized using the contrast limited adaptive histogram equalization algorithm (CLAHE) [32], as shown by the authors in [21]. Finally, side-scan sonar and all other navigation and control log data can be exported from Neptus in many formats: .kml, .mat, .csv, just to name a few.

AUV localization precision is on the order of 0.5–1 m, since a now very precise GPS sensor gives the upper bound on localization precision. When underwater, the localization uncertainty increases significantly as dead reckoning only accumulates error, but is corrected by the DVL as soon as it obtains the bottom lock

*4.3. Visual Data*

Visual surveys of some of the UCH sites and their surroundings both underwater (by BlueROV and/or AUV Lupis) and above sea surface (by DJI Phantom UAV). Camera onboard AUV Lupis has 1.4 MP and is fixed to capture things directly below the vehicle, so it cannot be used for the detailed full 3D modelling of a site. However, since it is synchronized with an external flash, it can be used for creating orthophotos or photomosaics. Images from AUV Lupis have timestamps and can be correlated with its position estimation data to further simplify photo alignment of photogrammetric software. BlueROV was used for capturing oblique HD images of some UCH sites and creation of 3D photogrammetric models. It does not have any underwater localization system, not even dead reckoning, so its images were fed "as is" into photogrammetric software after batch automatic brightness filtering in Photoshop. UAV DJI Phantom captures images in 20 MP resolution. It was used for capturing the shoreline in the vicinity of the UCH sites. It has a GPS sensor so it embeds it into metadata of the images that photogrammetric software can read later on. The precision of UAV's GPS is 1–2 m.

Agisoft Metashape was the software used for photogrammetric processing of images captures by all three vehicles. Photo alignment, point cloud generation and mesh processes were all set to high settings to get as detailed models as possible given the quality of the input images. The use of underwater robots for photogrammetric recording purposes was a use case to prove how effective these systems can be compare to diver photographers, especially at depths of more than 50 m. The quality of the models is graded rather subjectively, since there are no precise models which could be used as benchmarks.

## 5. Results

LABUST has always had a close partnership with maritime archaeologists. On one side, these interdisciplinary collaborations give the archaeologists the state-of-the art technology to survey the underwater sites, and on the other side, we as roboticists obtain application-oriented case studies for marine robotics control, path planning and sensor data processing algorithms. These collaborations include scanning the Senj (Croatia) fortress' well in 2007, inspecting the Ninnucia shipwreck (sunk in 1942) close to Rogoznica (Croatia) in 2008, surveying the Giuseppe Garibaldi shipwreck close to Cavtat (Croatia) in 2009, Roman villa rustica and the shipwreck Viribus Unitis close to Pula (Croatia) in 2010, side-scan sonar large-scal missions in search of new archaeological sites around Hvar island (Croatia) in 2010, inspecting the aquatorium of Kornati national park (Croatia) 2010–2012, work at Gnalić shipwreck (Gagliaana grossa from 1583) in 2011–2013, as well as the underwater archaeological site at Valgjärv (Estonia) in 2014.

This section presents the results of our more recent underwater archaeology endeavors. Section 5.1 brings the results of optoacoustic suvey of the ancient Caesare Maritima's harbour (Israel) done in 2014. Underwater archaeological sites surveyed in the scope of BLUEMED project (2016–2019) are given in the following subsections, namely Underwater archaeological park of Baiae (Italy) in Section 5.2, amphorae cage and ancient dolii sites close to Cavtat (Croatia) in Section 5.3, four ancient Greek shipwreck from the Western Pegaseticos in Section 5.4. Finally, the survey results of S.M.S. Szent Istvan shipwreck (sunk in 1918) made in 2019 are presented in Section 5.5. Parts of the results presented in this Section are published by the authors in [2,4,5,33].

### 5.1. Caesarea Maritima

In the last decade of the first century BCE, the building of Caesarea Maritima (shown on a map in Figure 6) was a bold effort in line with the prestige of its ruler, Herod I of Judea, and the influence of its founder, the Roman emperor Caesar Augustus. The word Caesarea comes from the Caesars family identity. The most prominent characteristic of the new city was a large artificial harbor occupying an region of about 200,000 square meters [34]. It involved large artificial breakwaters built from hydraulic cement and brought up from the seafloor. The ruins of Caesarea are the core of a national park adjacent to Qesarya's modern city. The sunken remains of the breakwaters and quays are covered in sand and scattered far apart, providing researchers with a daunting challenge attempting to recreate Herod's initial scheme.

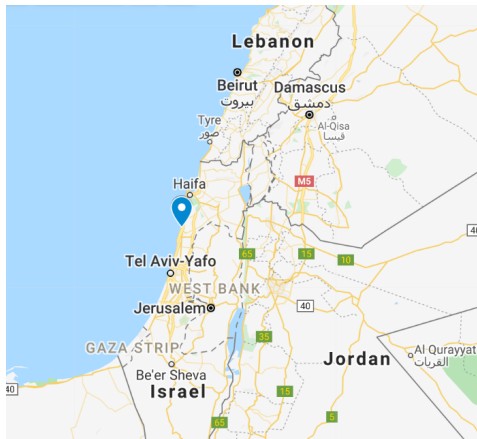

**Figure 6.** The location of the Caesarea Maritima underwater archaeological site.

At Caesarea, three modern techniques have recently been introduced with inconsistent yet nevertheless important results: multibeam, sub-bottom sampling, and magnetic sampling [35]. The promise of the two latter technologies is their ability to record features below the visible site's highly disturbed surface which is essential for understanding the original structures.

### 5.1.1. Methodology

The autonomous surface vehicle PlaDyPos built at LABUST started the first fused multibeam and photographic images of the sunken port systems in the summer of 2014. The purpose was to construct Caesarea's first full-scale, fully georeferenced underwater site map with a degree of precision and information usually seen only in small-scale underwater excavations. The field studies in PlaDyPos concentrated on parts of both Caesarea Maritima's inner and outer Herodian harbours. Moreover, the ruins of a Roman pier at neighboring Sdot Yam were mapped to the south.

Two types of data were gathered during the trials at Caesarea Maritima: a georeferenced point cloud of the seabed and archaeological features using the Norbit MBES, and visual imagery using a low-light mono camera, the Bosch FLEXIDOME IP starlight 7000 VR, in a custom waterproof enclosure. A GoPro Hero3 camera was also mounted onto the vehicle in a protective housing to capture extra footage for many tasks. It acquired the georeferenced point cloud by conducting pre-programmed lawnmower missions around the site area.

### 5.1.2. Results

The first objective was a survey of the base of a Roman Round Tower and Crusader Square Tower. Once replicated as a digital 2.5D image (Figure 7a), the sand and debris turn into identifiable architecture. The findings are ideal for visualization of GIS, such as utilizing Google Earth as seen in Figure 8a.

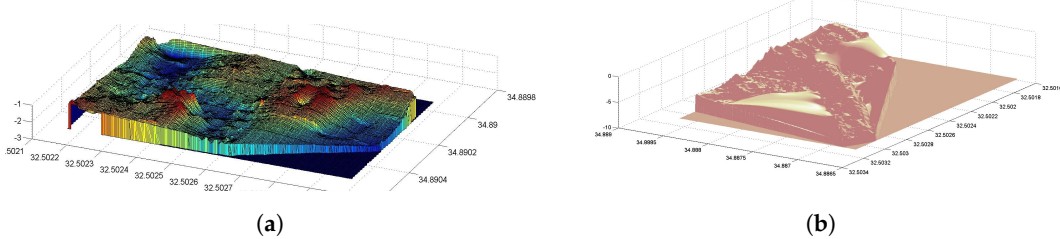

|  (a)  |  (b)  |

**Figure 7.** (**a**) 2.5D visualization of a Roman and Crusader towers foundations at Caesarea [2]. Based on multibeam echosounder/sonar (MBES) data collected by the ASV PlaDyPos. (**b**) Digital reconstruction of a 250 m section of Herod's southern breakwater [2]. Based on MBES data collected by the ASV PlaDyPos.

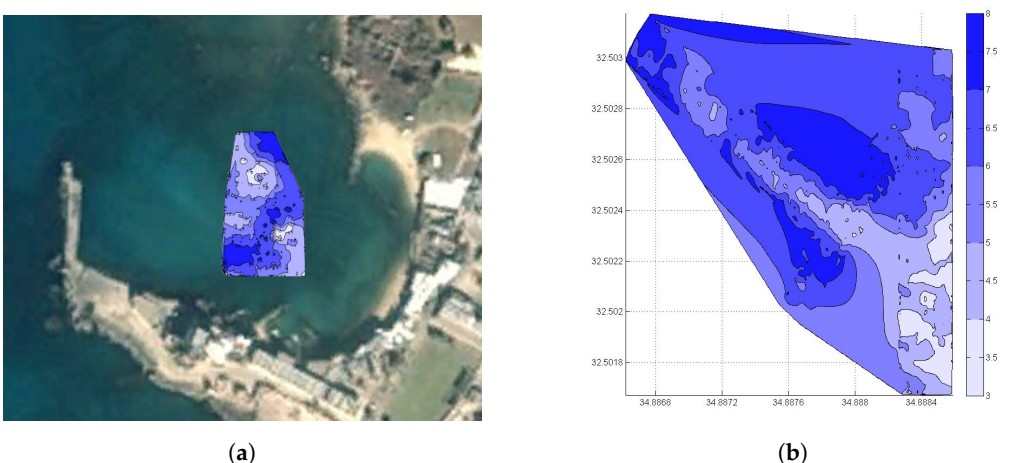

|  (a)  |  (b)  |

**Figure 8.** (**a**) Google Earth image of Caesarea's inner harbor with an overlaid bathymetry map [2]. Based on MBES data collected by the ASV PlaDyPos. (**b**) Partial bathymetry map of the southern breakwater in Caesarea's outer harbor [2]. Based on MBES data collected by the ASV PlaDyPos.

Herod's outer harbor became more open and deeper, with the area being studied reaching a depth range of 3–8 m. Three surveys were carried out over a distance of 250 m of the collapsed

southern breakwater and the findings were compiled to create a 2.5D reconstruction (Figure 7b) and a microbathymetry map (Figure 8a).

The optical data were used to generate offline a photomosaic of the region being surveyed. We validated freely available tools such as Microsoft ICE for picture stitching, as well as in-house applications built at LABUST. Image stitching software uses only the optical data, so the mosaics generated in subsequent processing must be matched with the telemetry data. LABUST has built tools for fusing both picture stitching and georeferencing optical and telemetry images. Figure 9a displays a photomosaic of one of the GIS (Google Earth) overlays of a mission transect. The degree of detail is shown in Figure 9b, showing a close-up of the region depicted in Figure 9a.

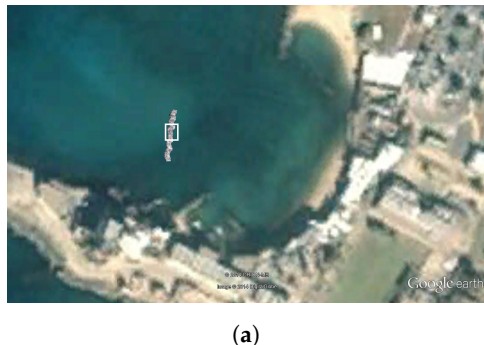 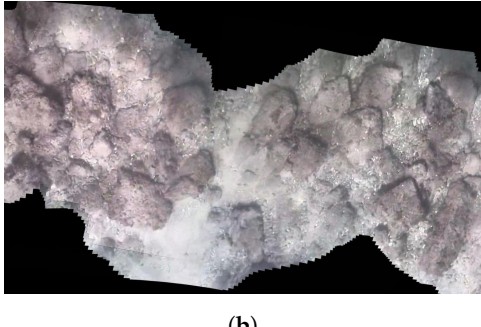

(**a**)  (**b**)

**Figure 9.** Photomosaic from Caesarea's inner harbor. (**a**) Mosaic overlaid on Google Earth (**b**) Detail of the area enclosed by the white box in (**a**) [2]. Based on visual data collected by the ASV PlaDyPos.

### 5.1.3. Discussion

The precision of the navigation data at Caesarea Maritima was in the order of 10 cm. Since the beam of the MBES is around $1°$, this means that its minimum trace on the seafloor is $0.01h$ at the depth $h$. This means that for the depths of Caesarea ancient port in the range 3–8 m, the best resolution of sonar data was 3–8 cm. The interpolation grid size in QPS Qimera was $10 \times 10$ cm, since higher resolution was not required by the archaeologists. The main issue with the documentation procedure of this UCH site is the fact that this marine environment is very shallow and dynamic. It is easily possible that each storm ruffles the sediment, thus changing the bathymetry of the area, covering previously uncovered walls and artefacts which were once exposed at the seafloor, while uncovering others. It would be thus useful to use a sub-bottom profiler which would penetrate the sediment and the underlying structures to uncover the true state and shape of this buried underwater UCH.

### 5.2. Baiae Site

Baiae, ancient city of Campania, Italy, is located on the west coast of the Puteoli Gulf (Pozzuoli) and lies 16 km west of Naples, as shown in Figure 10. Baiae was called after Ulysses' helmsman Baios according to custom. Because of its curative sulfur springs, the city is known as Aquae Cumanae in 178 BC. The mild climate of Baiae, the thermal springs and luxuriant vegetation made it a popular resort during the Roman Republic's later years. Many splendid villas, including those of Julius Caesar and Nero, were established at Baiae. Owing to nearby seismic activity (bradyseism), more than 100 m of the ancient site is now underwater in the harbor.

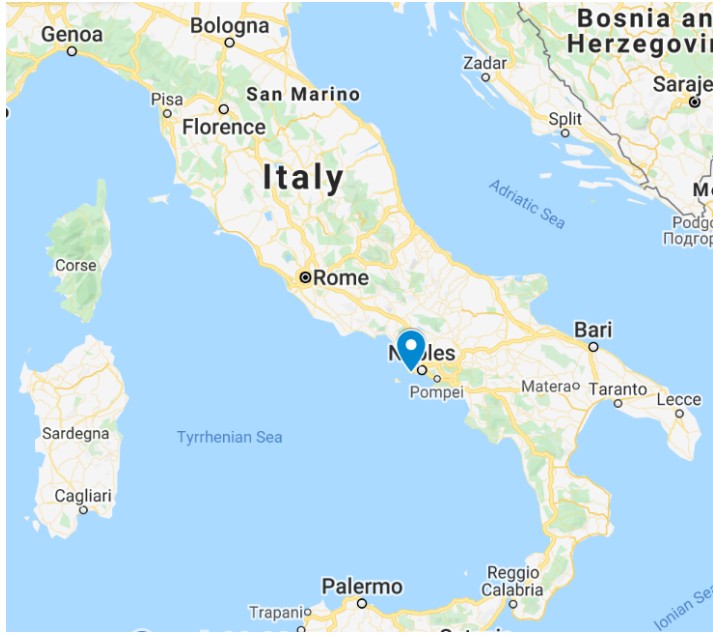

**Figure 10.** The location of the Baiae underwater archaeological site.

5.2.1. Methodology

In 2018, our team went to Baiae to record the underwater archaeological site at depths of 6–8 m. This was done under Interreg Mediterranean co-funded project BLUEMED. Survey operations at Baiae pilot site were conducted by our team on 14-18.5.2018. The main emphasis of this operation was to gather as much bathymetric data of the site as possible, so the ASV PlaDyBath was mostly deployed for survey missions. For this reason, the ROV has not been used for visual inspection of the pilot site, and UAV was not used, since we did not have an official permission for operating it in the Baia port area. The boundary of the survey area has been determined based on a GIS map provided by the local diving center. Detailed survey missions were planned based on the georeferenced mosaic of side-scan sonar imagery recorded by LAUV Lupis, see Figure 11.

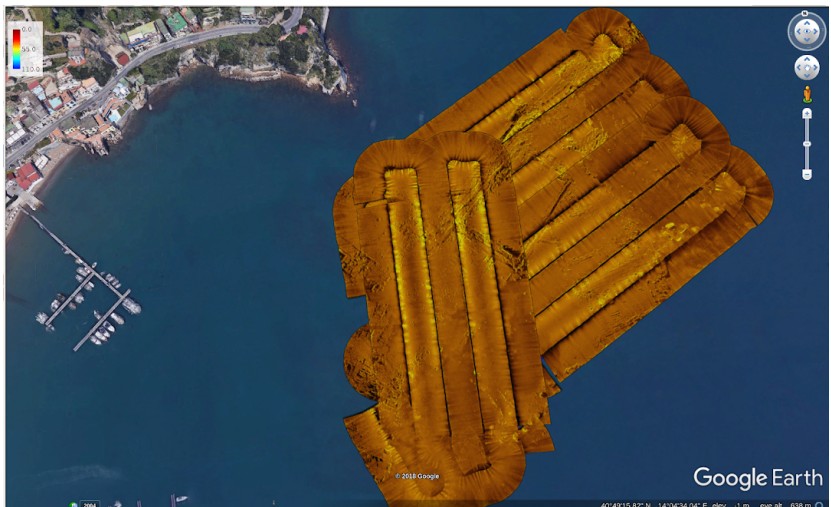

**Figure 11.** Side-scan sonar imagery mosaic recorded by LAUV Lupis.

A boat was rented from the local diving center for the bathymetry and photogrammetry data collecting operations by the surface and underwater autonomous marine vehicles. It was quite spacious, so deployment and recovery of the autonomous vehicles by the crane were not problematic. The autonomous surface vehicle (ASV) PlaDyBath's survey path was designed lawnmower-shaped

survey missions along and across the area of interest in the Baiae bay. The missions were planned with 90–120° field of view angle of the Norbit multibeam sonar used for bathymetry, having in mind to cover the whole area with complete overlap between any two adjacent along-track survey lines, and having across-track survey lines to maximize the amount and quality of the bathymetry data as much as possible, and to avoid holes in the bathymetry map. As soon as one mission would finish, the sonar data and position/attitude data were transferred from the ASV to a laptop, and bathymetry data were processed. Meanwhile, another mission was started, so data collection and processing were parallelized as much as possible.

### 5.2.2. Results

The georeferenced side-scan sonar mosaic shown of the Baiae underwater archaeologycal park is shown in Figure 11. Results of multibeam sonar data postprocessing in QPS Qimera software are shown in Figure 12. Bathymetry map of the whole survey Baiae area is given in Figure 12a, with details of the sunken Vila dei Pisoni in Figure 12b, and details of the sunken Vila Protiro in Figure 12c. It is notable that the walls of these villas can clearly be distinguished from the surrounding seafloor.

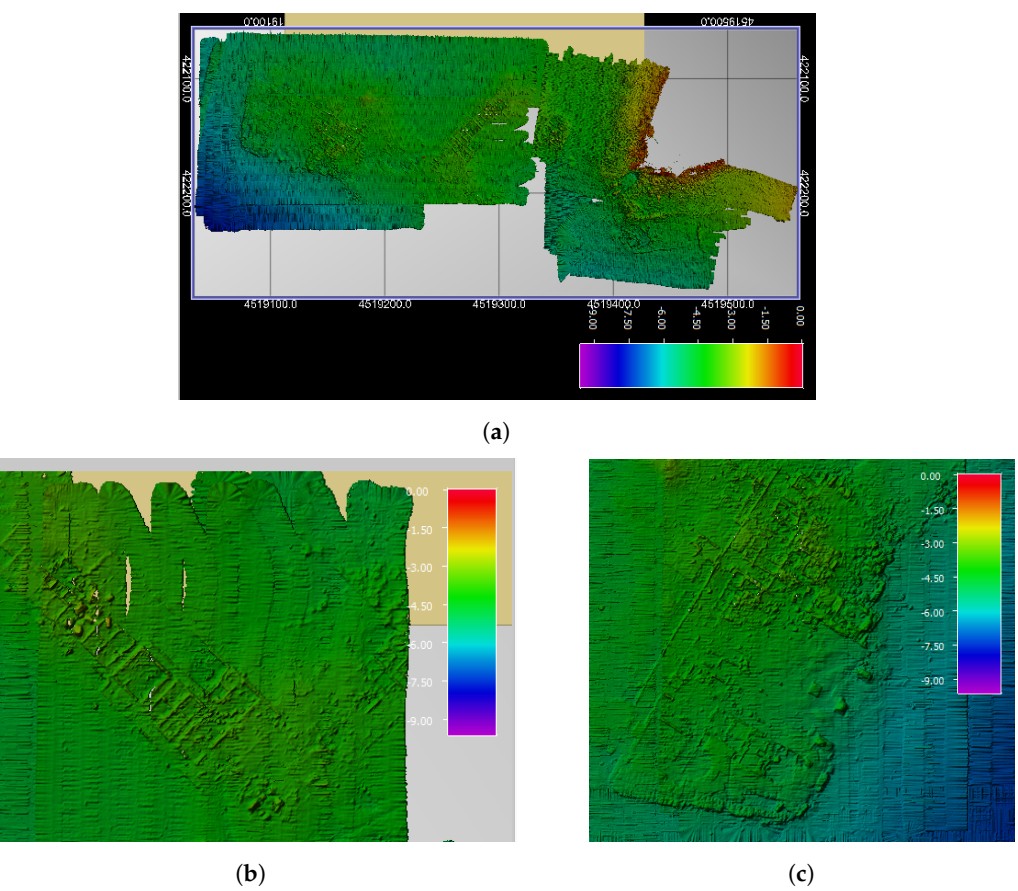

**Figure 12.** (**a**) Results of the whole Baia pilot site's bathymetry. (**b**), Details of the submerged Villa dei Pisoni. (**c**) Details of the submerged Villa Protiro. 2.5 bathymetric models are based on MBES data collected by the ASV PlaDyBath.

### 5.2.3. Discussion

Surveys at Baiae site were hazardous for the ASV close to the steep cliff of the cape on the north-eastern corner of the UCH area, as well as due to the busy marina just north-west of the site. Even though the UCH area is delimited with bouys, many vessel do not abide and sail right through the marine protected area. Moreover, the AUV operations were done from the surface with more dense

lawnmower survey missions, because of potential safety risk and possible collisions with other vessels during the resurfacing.

The precision of position in bathymetric model of Baiae site is on the order of 10 cm because there we were not given access to the local base stations for georeferencing corrections through NTrip client. Since the beam of the MBES is around 1°, this means that its minimum trace on the seafloor is $0.01h$ at the depth $h$. This means that for the depths of ancient Roman Baiae site in the range 6–8 m, the best resolution of sonar data was 6–8 cm. The interpolation grid size in QPS Qimera was $10 \times 10$ cm, since higher resolution was not required by the archaeologists, and all the walls and masonry details of villas are visible in bathymetric maps. In the initial bathymetric interpolated model, there were waves which translated from the surface to the sonar data. This is a consequence of the INS system not having a heave sensor. However, Qimera's Wobble Analysis tool and further spline interpolation solved this problem in the post-processing phase.

One of the most important wider social impacts our survey work had on the local Baiae community was the deisgn of DryDive mobile application, which our colleagues from University of Calabria made using, in part, our recorded data in the scope of Horizon 2020 i-MARECULTURE project.

*5.3. Cavtat Sites*

The Adriatic Sea is teeming with undiscovered ancient shipwrecks and untold treasures. Two remarkable underwater sites in Croatia, one with around 700 amphorae and the other one with dolii are located in front of Cavtat, a town south of Dubrovnik, as shown in Figure 13.

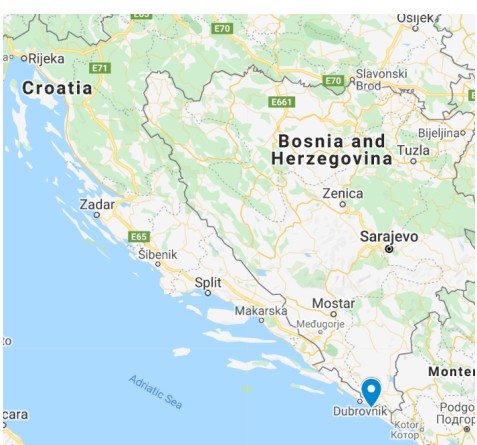

**Figure 13.** The location of the Cavtat underwater archaeological site.

5.3.1. Merchant—Ancient Greek Sailing Boat with Dolii

One special feature, though, is the underwater site officially discovered in 1996, representing the only preserved site of large ancient ceramic vessels for cargo-dolia (Greek: Pithos, Latin: dolium) on the eastern Adriatic coast. It is an intact location at the depth of around 20 m. The site covers $10 \times 20$ m area. Traces of the sunken boat have not been noted, though dolia's discovery may simply illustrate the unproven shipwreck. Throughout the 2nd and 1st centuries B.C., dolia was also found on Roman warships in the Antiquity. These became the primary way of shipping goods in the 1st century, and the features of ancient tankers were taken over by vessels that transported them. Their capacities ranged between 1500 and 3000 L. Cavtat's dolia dates back to the 1st century and its estimated capacity ranged from 1200 to 1400 L.

5.3.2. Amphorae Cage

An ancient Greek shipwreck believed to hold a cargo worth around \$5–8 million in today's value has recently been protected with a large cage to shield the ship's cargo from scuba looters who would be tempted to steal the objects. The wreck of these 700 vases was a Greek trade vessel of the

second century bearing a shipment of earthenware amphoras of olive oil and wine, which sunk just off the coast of a small town of Cavtat, 20 km south of Dubrovnik, Croatia. In 1999, Boris Obradovic, the director of a nearby Scuba Diving center discovered the wreck and now guides more seasoned scuba divers down to the debris. The wooden ship is almost entirely decomposed, but oddly enough, the ceramic amphora is holding the wine and olive oil, which are still intact and lined in the ships' holds row after row. This freight that has considerable historical importance and interest on the black market is worth quite a bit of money, prompting the Croatian authorities to cover the wreck with a huge heavy duty metal frame. The cage itself is about 20 m long and 10 m high, and has a big hinged door that can be locked closed.

### 5.3.3. Methodology

In 2018, LABUST team went to Cavtat to record the dolia and amphorae cage sites, which are only a few hundred meters apart. This was done under Interreg Mediterranean co-funded project BLUEMED. The team worked from a catamaran work boat anchored close to the underwater archaeological sites. The deployment of the bulky ASV was simplified by using a mobile manual crane operated from the spacious flat stern of the catmaran work boat. This enabled the multibeam sonar mounted ASV to perform survey missions and record bathymetric data more easily, thus losing less time during the battery changes.

BLUEROV2 was used for visual data collection, mainly at the dolia site. Easily deployable, and manually controlled, ROV has been shown to be very useful, especially since there is a direct visual link through the tether back to the operator's screen.

One of the goals of this Cavtat trial was to gather dataset for photogrammetry of the Supetar islet close to these two underwater archaeological sites. The UAV took HD photos of the island in a crosshatch coverage plan, with 70% across- and along-track overlap between adjacent photos, at an altitude of 40 m for safety reasons.

### 5.3.4. Results

Precise positions of both sites were determined from the georeferenced side-scan sonar imagery of the area, recorded by the LAUV Lupis, as shown in Figure 14. The resulting bathymetric map of the area is given in Figure 15.

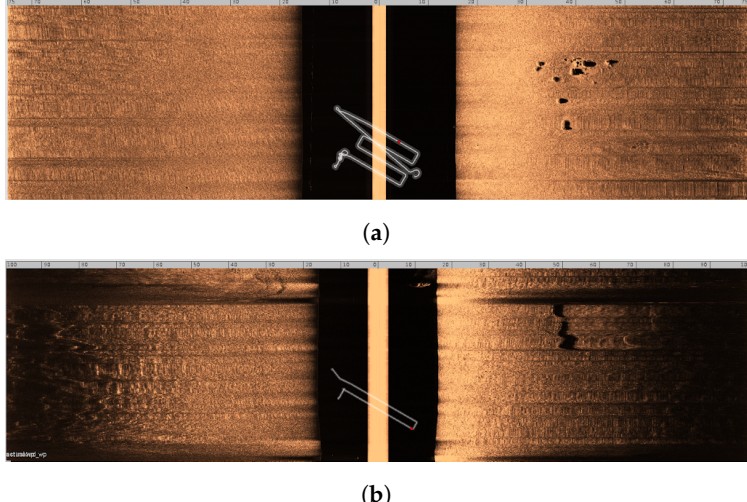

(**a**)

(**b**)

**Figure 14.** Side-scan sonar imagery of the sites. (**a**) Dolii site. (**b**) Amphorae cage site.

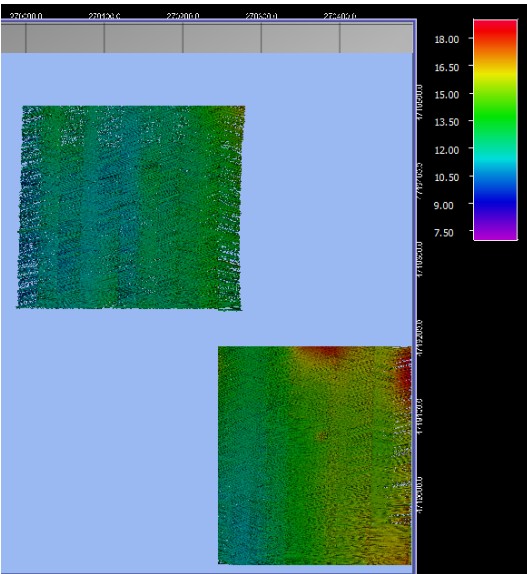

**Figure 15.** Bathymetry map of the underwater archeological sites in front of Cavtat, Croatia. Moreover, the 2.5 bathymetric model is based on MBES data collected by the ASV PlaDyBath.

In the post-processing phase, a 3D model of one dolium was generated from downsampling a 25 fps HD video from ROV's logs, as shown in Figure 16b. The biggest problem with the use of frames extracted from the recorded video is motion blur, which renders 3D models of relatively less quality compared to the ones generated from still photographs of a DSLR camera with an external high-power flash. Nonetheless, this opens a possibility to document an underwater archaeological site much faster and with much less logistics compared to the case when divers record the site. The result of the photogrammetry post-processing of the Supetar islet aerial photos is given in Figure 17b.

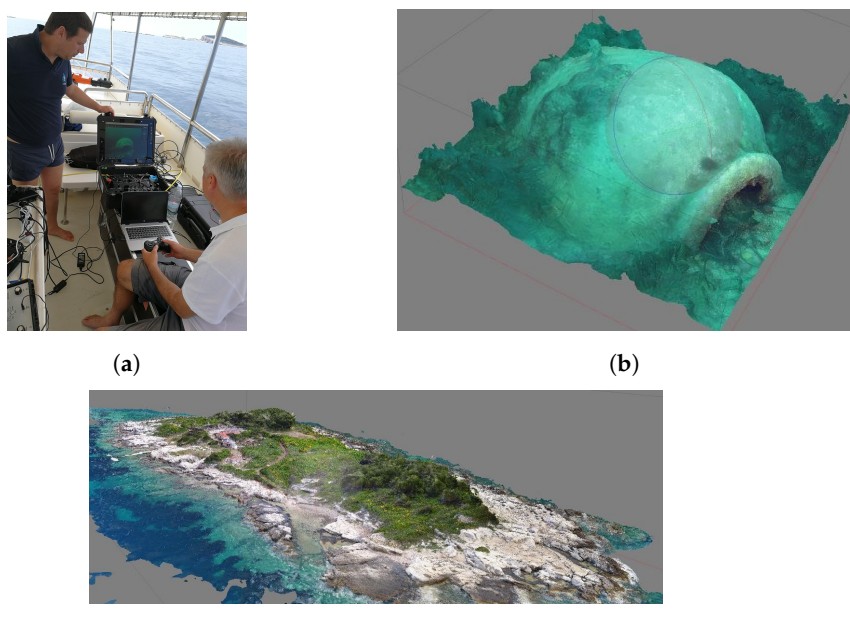

**Figure 16.** (**a**) Operating BLUEROV2 at dolia site in Cavtat, Croatia, with a direct HD video feedback on a high-contrast screen.(**b**) Photogrammetric 3D reconstruction of a dolium based on frames from remotely operated vehicles (ROV)'s camera. (**c**) 3D reconstruction of the island Supetar in front of Cavtat bay, close to the diving locations of the amphorae cage and dolia locations.

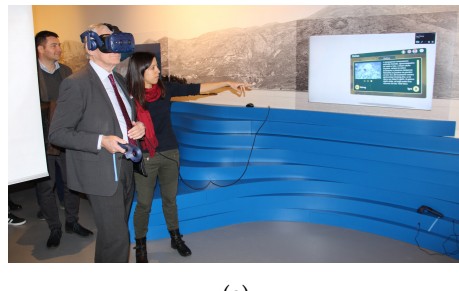 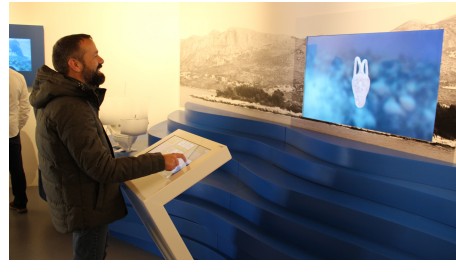

|(**a**)|(**b**)|

**Figure 17.** Knowledge Awareness Center opened in Konavle near Cavtat, Croatia. (**a**) Presentation of the so called underwater cultural heritage sites (UCH) site dry dive experience through virtual reality. (**b**) Interactive content presentation, e.g., 3D reconstructed model of an amphora at Cavtat Amphorae cage site that offers information on a 4K ultra HD TV and interaction over a touchscreen.

### 5.3.5. Discussion

The precision of the navigation data at Cavtat site was on the order of 1 cm, since corrections w.r.t. the local CROPOS base station system were available through NTrip client connected over a 4G modem onboard the ASV. Since the beam of the MBES is around 1°, this means that its minimum trace on the seafloor is $0.01h$ at the depth $h$. This means that for the depths of the amphorae cage and dolii site in the range 20–25 m, the best resolution of sonar data was 0.2 m. The interpolation grid size in QPS Qimera was $20 \times 20$ cm, since higher resolution was not required by the archaeologists, and all the walls and masonry details of villas are visible in bathymetric maps.

One of the most important results of our surveys at Cavtat UCH sites was the opening of a Knowledge Awareness Center (KAC) in Konavle near Cavtat in January 2020. This was done in the scope of BLUEMED project, as shown in Figure 17. There, our findings were presented in various forms. One form of knowledge presentation was a virtual reality (VR) set that was build based on our bathymetric model and merged with high-resolution photogrammetric model done by our colleagues from University of Calabria led by Prof. Fabio Bruno.

### 5.4. Western Pagaseticos Sites

Four underwater archaeological sites were surveyed by our autonomous vehicles in the Westerns Pagaseticos, Greece, a map of which is shown in Figure 18.

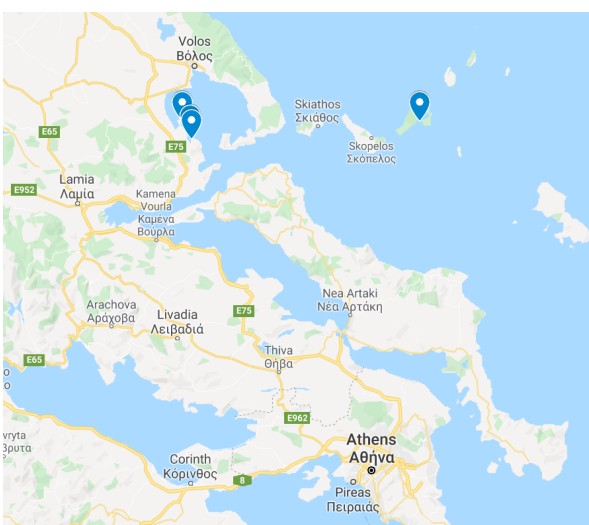

**Figure 18.** The locations of the underwater archaeological sites in Western Pagaseticos.

### 5.4.1. Alonissos—Peristera Shipwreck

This shipwreck was found in the early 1990s. The uncovered pieces of the wooden ruin of Peristera have been rotting away for a long time, but the remaining cargo has a fantastic 4000 amphorae seascape. The ship, claimed by historians to be a huge Athenian barge holding wine-filled amphoras, presumably sunk at the end of the fifth century B.C. The hull still lies submerged at around 30 m, and archaeologists believe it is the biggest ship of its type found beneath the sea.

At this site, the team first obtained a georeferenced mosaic of the collected side-scan sonar data collected by the AUV Lupis around the pilot site given in Figure 19a. The ASV was used to gather bathymetric data of the pilot site in more detail, with a narrow angle of view of the sonar, and high percentage of swath overlap between neighbouring survey lawnmower lanes. A wider sonar angle of view, but again with a high percentage of overlap between consecutive survey lanes, were used to gather bathymetric data of the pilot site's surrounding area. The overlay of all survey missions planned, as well as wide areas, detailing the bathymetry map of the pilot site itself are given in Figure 19b.

UAV DJI Phantom 4 UAV was used to gather photos of the Peristera island part just in front of the pilot site. It was programmed to execute a crosshatch mission at altitude of 40 m, and a 70% along- and across-track overlap between photos. Results of the 3D reconstruction of the Peristera island part are given in Figure 19c, based on 381 photos.

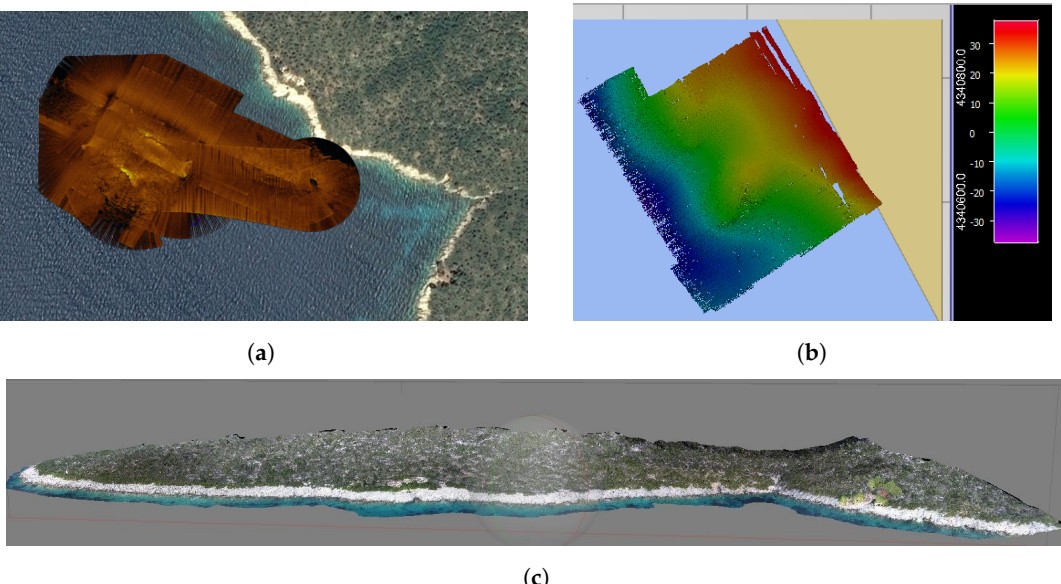

(**a**)                    (**b**)

(**c**)

**Figure 19.** (**a**) Side-scan sonar mosaic of the Peristera site georeferenced and shown in Google Earth, based on SSS data collected by LAUV Lupis. (**b**) Results of bathymetry of the area surrounding the Peristera pilot based on MBES data collected by ASV PlaDyBath. (**c**) 3D reconstruction of the part of Peristera island in front of the pilot site locations, based on visual data collected by UAV DJI Phantom 4.

Merging of the low-resolution acoustic model of the seafloor with the high-resolution textured photogrammetry model is shown in Figure 20, which is one of the results stemming from the aforementioned BLUEMED project and collaboration with our partners from University of Calabria led by Fabio Bruno [4,5]. The reason why this merger was performed is that one of the BLUEMED project goals was to preserve the underwater cultural heritage sites in a digital form, so that visitors of the museum in the vicinity of the site can experience the diving visit to the site by wearing a virtual reality set. This allowed for the low-resulotion acoustic model of the area to provide the georeferenced collocation to the high-resolution textured photogrametric model of the site, but also the general features for the UCH's surrounding area.

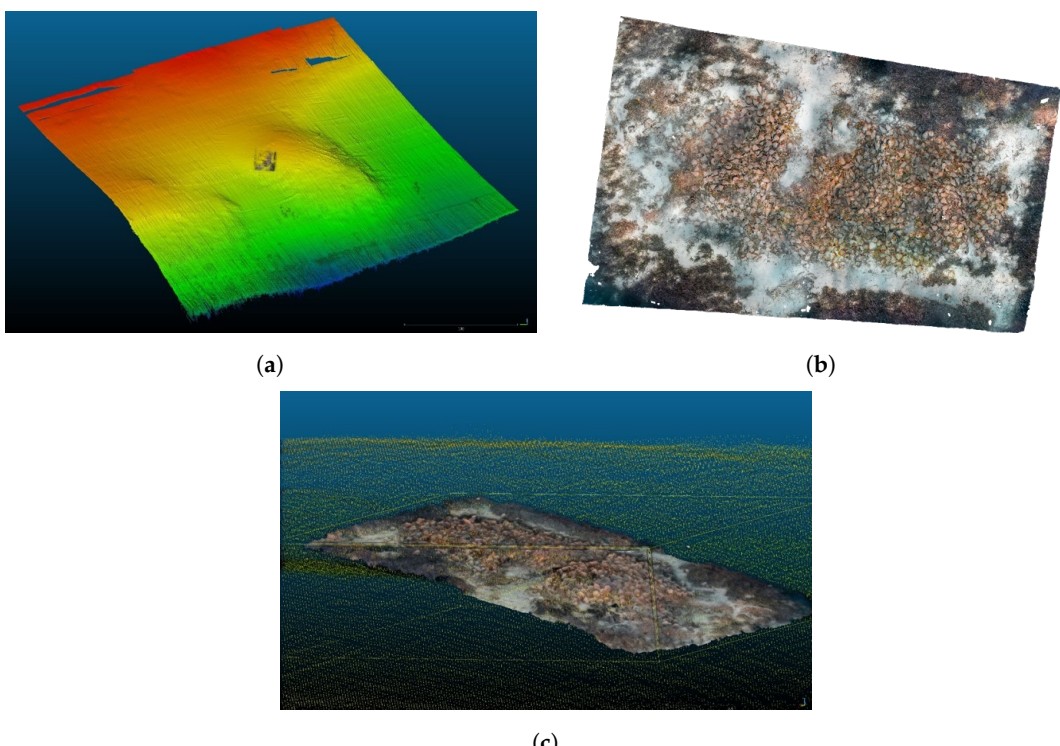

(**a**)  (**b**)

(**c**)

**Figure 20.** (**a**) Bathymetry of the area of the site based on MBES data collected by ASV PlaDyBath. Small box in the middle is the collocated 3D photogrammetric model of the Peristera shipwreck site. (**b**) Orthophoto projection of the photogrammetric 3D model of the site based on high resolution photos taken by divers. Courtesy of Fabio Bruno. (**c**) Merge of the acoustic and optical 3D models [4,5]. Courtesy of Fabio Bruno.

### 5.4.2. Kikinthos Shipwreck—West Pagasetic Gulf

Kikinthos islet is a small breakwater located east of the Bay of Amaliapolis. In ancient times, Kikinthos was used as a quarantine for the seamen who had returned from their voyages to Amaliapolis. The shipwreck was discovered under the auspices of the Hellenic Institute of Marine Archaeology and the guidance of the maritime archeologist Elias Spondylis during an underwater survey undertaken in 2005 at the northwestern end of Kikinthos Islet. The remains of a primarily pithoi Byzantine shipwreck cargo (large shipping containers) are found about 3–11 m from the shore. Large pithos fragments, which can be traced to at least three separate styles, occupy an area of about 8 × 6 m. There are also fragments of two types of amphorae, dated from the 12th–13th centuries AD, in the pithoi. The styles of pithoi are traced to the 8th–9th centuries AD, but it seems that they coexist with the later amphorae, as storage vessels were typically used for lengthy periods of time.

Bathymetry of the shipwreck and the whole inner side of the Kikinthos islet facing towards Amaliapolis is shown in Figure 21a. DJI Phantom 4 UAV was used to gather photos of the site's surroundings. On these pilot sites, it was flown slowly in manual mode at an altitude of approx. 10–15 m, to ensure a high percentage of photos overlapping. The camera was oriented directly towards the shore, to obtain more details of the shore. Photogrammetrics 3D reconstruction of the whole Kikinthos island is shown in two parts: side facing towards Amaliapolis (see Figure 21c, and the other side facing out (see Figure 21c)).

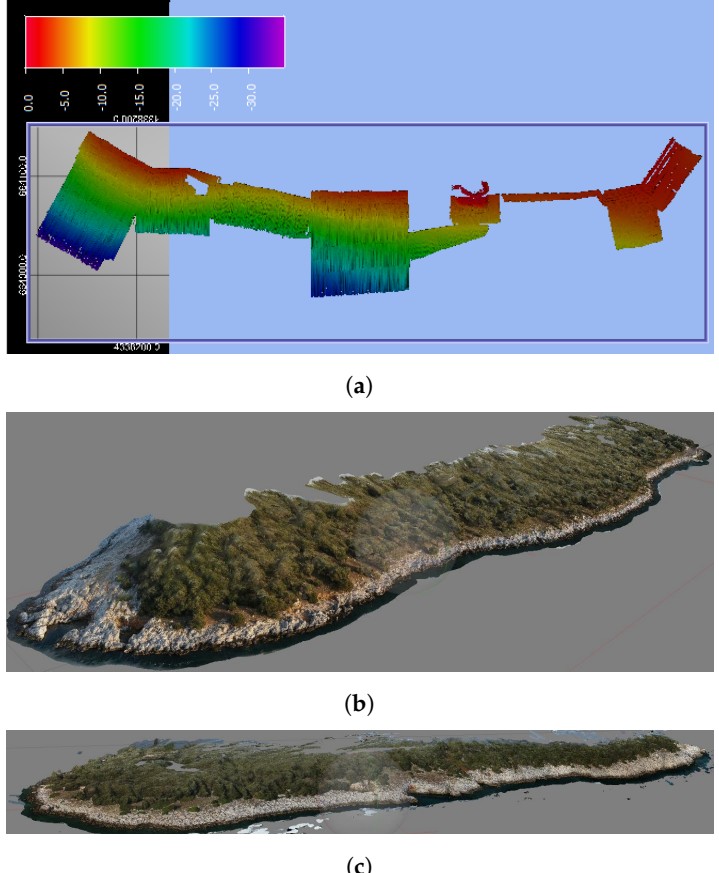

**Figure 21.** (**a**) Results of bathymetry of the area surrounding the Kikinthos site based on MBES data collected by ASV PlaDyBath. (**b**,**c**) Three-dimensional reconstruction of the Kikinthos island in front of the pilot site locations, facing towards Amaliapolis and the outer islet side, respectively. Based on images taken by UAV DJI Phantom 4.

### 5.4.3. Akra—Glaros Shipwreck

This archaeological underwater site is located in an area opposite Nies, a coastal village in Magnesia Prefecture and near Amaliapolis City. According to the Hellenic Institute of Marine Archeology, which investigates the area from 2000 to present under the direction of the marine archeologist Elias Spondylis, at least four shipwrecks were recognized: the Hellenistic one (3rd–2nd century BC), the Early Roman one (1st–2nd century AD) and two of the Middle and Late Byzantines (12th–13th century AD), where late Roman pottery is also present. The reports related to the above shipwrecks are so scattered and confused that the classification of the different shipwrecks is an activity that is very challenging and not yet complete.

Bathymetry of the shipwreck and wide area around the underwater archaeologycal site is shown in Figure 22a. Photogrammetrics 3D reconstruction of the whole Glaros cape shore is shown in Figure 22b.

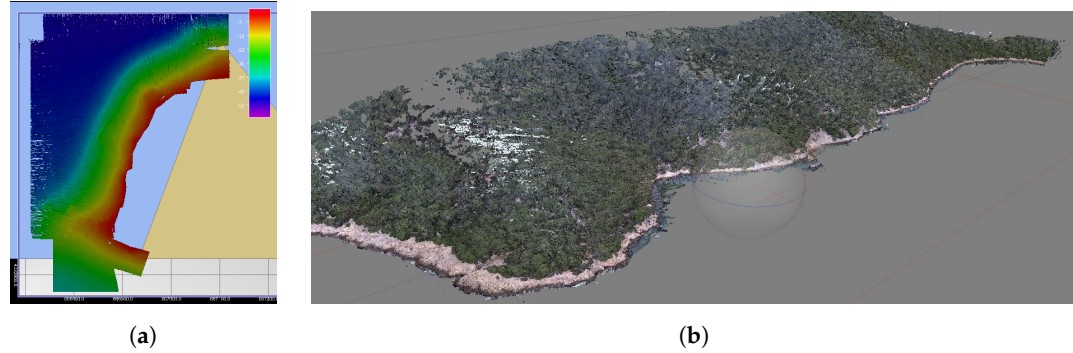

(**a**) (**b**)

**Figure 22.** (**a**) Results of bathymetry of the area surrounding the Glaros site based on MBES data collected by ASV PlaDyBath. (**b**) 3D reconstruction of the Glaros cape in front of the pilot site locations. Based on images taken by UAV DJI Phantom 4.

### 5.4.4. Telegrafos Shipwreck

Telegrafos Bay is situated in Magnesia Prefecture and near the town of Amaliapolis. The shipwreck was first discovered in 2000 by the Hellenic Institute of Marine Archeology team that explored the region of the south-western coast of the Pagasetic Gulf, and was then thoroughly excavated from 2003 to 2008, also by HIMA, under the supervision of the marine archaeologist Elias Spondylis. Unfortunately, previous to the excavation, the site was robbed. Nonetheless, the discovery brought the majority of the cargo to light, and a detailed analysis contributed to the identification of three major forms of Late Roman (4th century AD) amphorae for the major cargo which could be traced to northern Peloponnesian (Corinth) and Eastern Aegean (Samos) roots. Facts show a ship sailing down the Late Roman sea routes—the recently formed Byzantine Empire through the Aegean Sea and down its edges.

Bathymetry of the shipwreck and wide area around the underwater archaeological site are shown in Figure 23a. Photogrammetric 3D reconstruction of the whole Telegraphos cape shore is shown in Figure 23b.

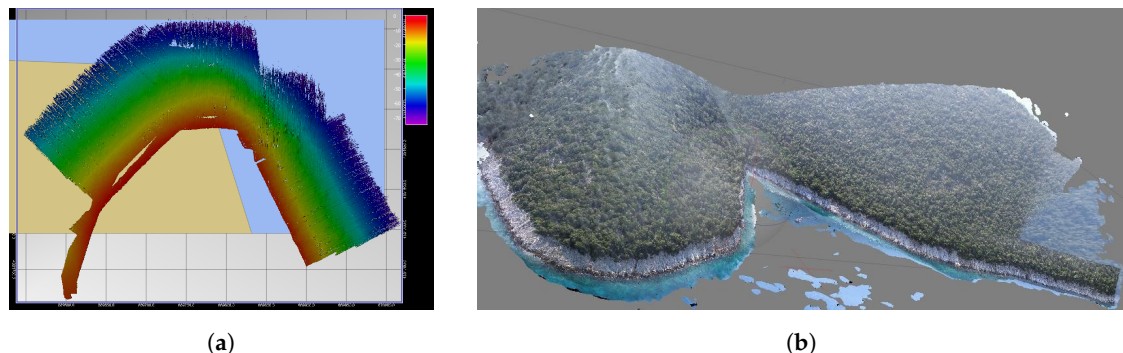

(**a**) (**b**)

**Figure 23.** (**a**) Results of bathymetry of the area surrounding the Telegrafos site. (**b**) Three-dimensional reconstruction of the Glaros cape in front of the pilot site locations.

### 5.4.5. Discussion

The precision of the navigation data at all four Wester Pagaseticos UCH sites was on the order of 10 cm, because corrections from local base stations were not available to us. Since the beam of the MBES is around $1°$, this means that its minimum trace on the seafloor is $0.01h$ at the depth $h$. At the Peristera, Glaros and Telegrafos UCH sites, the measured depth was on a range 0–60 m. The Peristera shipwreck is located in the area where the depth is 20 m. For these shallow parts of the surveyed area, a $20 \times 20$ cm interpolation grid size in QPS Qimera was used. For the deeper parts of the survey, the area grid cell was set to 30 cm, since they were needed only for general seafloor morphology when merging optoacoustical model of the shipwreck in virtual reality. The same applied to Glaros and

Telegrafos UCH sites, since there, the remaining artefacts after shipwrecks lie no deeper than 20 m. At the Kikinthos UCH site, the measured depth was in the range 0–30 m. Here, interpolation grid cell size of 20 × 20 cm was used for the whole surveyed area.

In the scope of the BLUEMED project, two additional KACs were opened in Greece. One was opened at the Island of Alnonissos on 11 September 2019 (see Figure 24) in the vicinity of the Peristera shipwreck. On 18 September 2019, another KAC was opened in Amaliapolis, Greece, just across the bay from the Island of Kikinthos, as shown in Figure 25. As in the case of Cavtat site KAC, both these KACs in Greece offer the visitors various type of educational content and the results of the research done in the scope of BLUEMED project. This includes posters, touchscreen monitors with interactive content, as well as virtual reality sets. Using these virtual reality sets, the visitors of these KACs can experience the so-called dry dive, i.e., to dive to the shipwreck UCH sites in virtual reality.

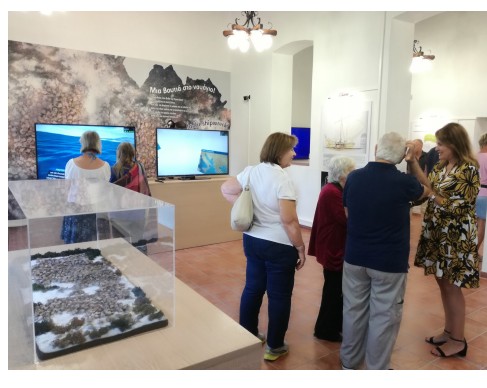
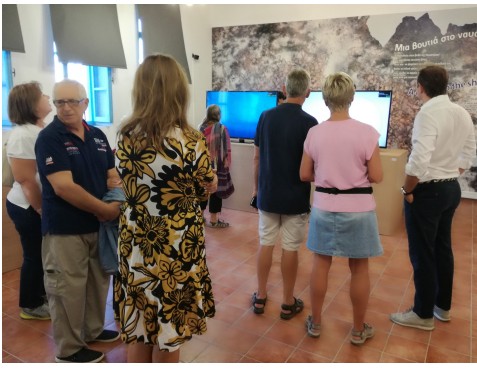

(**a**)                                          (**b**)

**Figure 24.** Knowledge Awareness Center opened at the Island of Alonissos, Greece. (**a**) Presentation of 3D cast model of the UCH site (left in the photo) based on optoacoustic model that was based on the bathymetric data presented here. (**b**) Visitors could learn a lot about the site either by reading posters on the walls of the KAC or by watching video content.

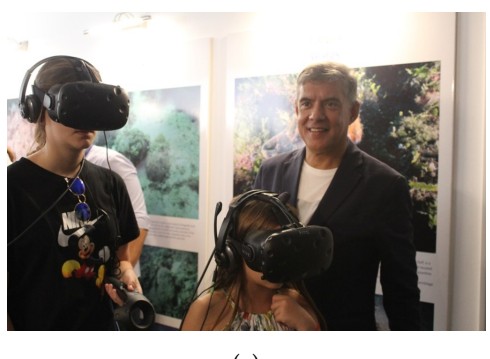
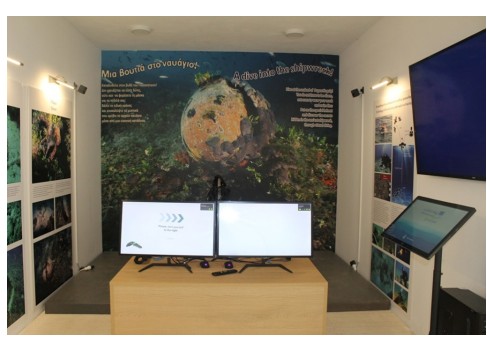

(**a**)                                          (**b**)

**Figure 25.** Knowledge Awareness Center opened in Amaliapolis, Volos, Greece. (**a**) Presentation of the so called UCH site dry dive experience through virtual reality. (**b**) Various posters and touchscreen TVs offering visitors to be fully immersed in the experience of the shipwreck's history but also the process of building its 3D optoacoustic model.

*5.5. Szent Istvan Shipwreck*

S.M.S. Szent Istvan, the only ship belonging to the Hungarian monarchy, met her end on 10 June 1918 shortly before dawn. It was sunk by Italian torpedo boats. On the 101st anniversary of this event, the shipwreck was recorded for the first time by our multibeam sonar-mounted autonomous surface vehicle. The shipwreck has already suffered irreversible degradation of her steel and iron hull. Thus, the main objective of the bathymetric surveys was to assess the current state of the shipwreck and to set up a foundation which future monitoring operations could be built upon and compared with. SMS Szent István was an Austro-Hungarian battleship of the Tegetthoff class, constructed in

Rijeka and Pula and completed in 1914. It was the only Austro-Hungarian ship to serve the Hungarian part of the monarchy [36]. Its blueprints are given in Figure 26a for reference in the remainder of this subsection.

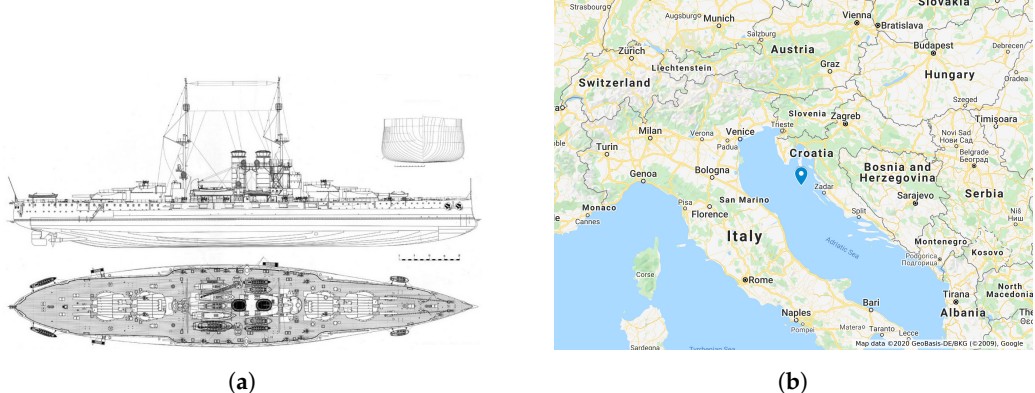

|     |     |
| :-: | :-: |
| (**a**) | (**b**) |

**Figure 26.** (**a**) Blueprint of the cross-section, side and top view of the SMS Szent Istvan [36]. (**b**) Location of the UCH site SMS Szent Istvan shipwreck.

The Szent Istvan wreck (deepest point at 68 m) has been visited so far by many local and foreign divers. It lies inverted with the deck facing the bottom, with the cannons still facing left. Drawings of the shipwreck by Danijel Frka are given in Figure 27. In Figure 27a, details of the southern side of the shipw's aft are shown, namely propellers, motor shafts, cannons, keel, as well as numerous fishing nets laying on the shipwreck and around it. A depression can be seen just above the cannons that was the consequence of the ship's hull imploding the air trapped inside during the sinking. Moreover, from the shadow below the southern side of the shipwreck's aft, it can be deduced that the shipwreck is leaning a bit to its northern side, thus creating an opening under the ship's hull on its southern side [36]. Details of the northern side of the torn bow, which broke during the sinking, are drawn in Figure 27b.

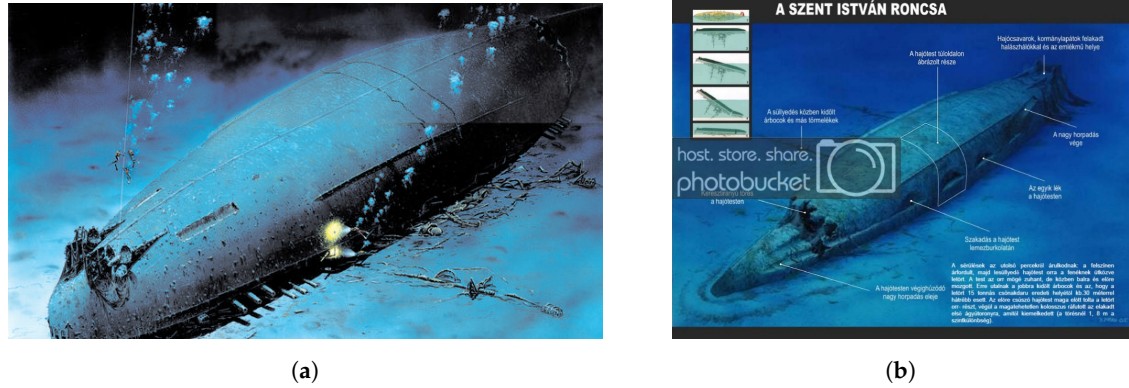

|     |     |
| :-: | :-: |
| (**a**) | (**b**) |

**Figure 27.** Drawing of the (**a**) aft (**b**) bow of the SMS Szent Istvan shipwreck. Image courtesy of Danijel Frka [36].

5.5.1. Methodology

As shown in Figure 28a, the shipwreck lays upside down on the floor, leaning on its superstructure and resulting in the south side of the hull been lifted from the seabed. The size of the gap on the stern part of the ship is shown in Figure 28b. Degradation of the steel hull underwater would eventually result in closing that gap until the hull completely collapses under its own weight. To measure the gap along the ship side, the profiling sonar carried by autonomous surface vehicle was used utilizing the methodology shown in Figure 28a. It was necessary to design a mission for the ASV PlaDyBath, which runs 50m from the south and north sides of the ship and parallel to the ship, as shown in

Figure 29 by yelow lines. The viewing angle of multibeam sonar was set to 60°, but the rays were tilted to the left by 15°, as shown conceptually in Figure 28a.

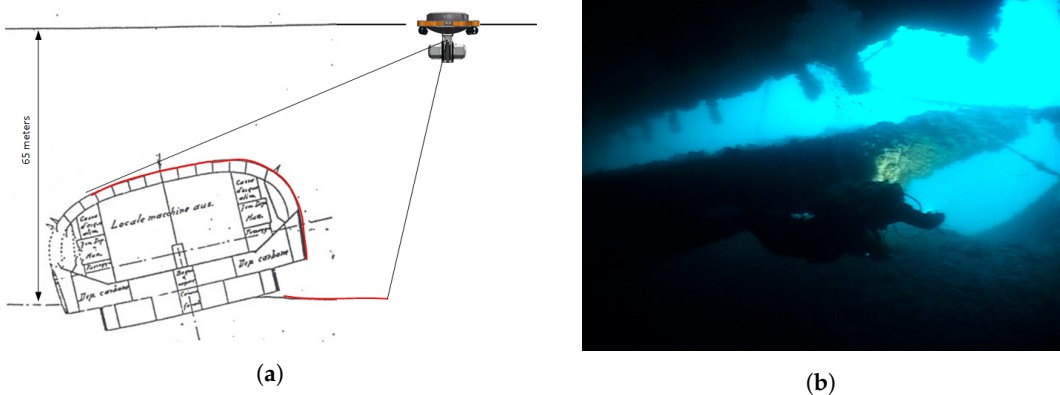

(**a**)                    (**b**)

**Figure 28.** (**a**) Methodology for measuring the Gap using ASV and the profiling multi-beam technology. Red line represents the sonified area. The image is conceptual; it does not respect the real proportions e.g., depth vs. hull size, nor sonar beam width and tilt angles. (**b**) Image representing the size of the gap between the ship hull and the seabed relative to diver. Image courtesy of Marino Brzac.

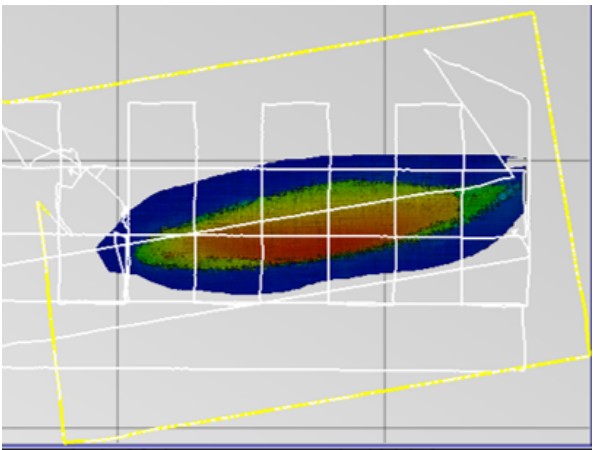

**Figure 29.** Overlay of ASV PlaDyBath's survey paths w.r.t. the top view of the bathymetric model of the shipwreck. (yellow) survey missions around the ship with tilted sonar beams to record the side of the ship (white) standard lawnmower missions along and across the wreck to capture the general morphology of the shipwreck.

5.5.2. Bathymetry Survey Results

An area of 200 × 75 m around the site was recorded by multibeam sonar, using standard lawnmower missions along, across, and from the sides of the wreck, as shown in Figure 29 by white lines. In missions planned along and accross the wreck to capture as much detail as possible, the sonar viewing angle was also set to 60°, but without ray tilting, and with adaptive ping frequency to ultimately obtain the highest quality shipwreck model from the sonar data. The navigational precision of the autonomous vehicle, and therefore the precision of geolocation of the 3D model reconstructed from sonar data, is of the order of 10 cm, which is more than sufficient for archaeological applications.

The missions planned in advance for the surface vehicle were in the form of transects spaced 25 m apart, thus providing sonar data with much redundancy and more detail in an important part of the shipwreck area. QPS Qimera software was used to reconstruct the bathymetric model from sonar and navigation data. A 0.5 m resolution was used for the bathymetric model because it is also realistically physically achievable accross-track resolution that the sonar mentioned above may have. Since each beam angle is approx. 1°, this would mean that the maximum physically achievable

resolution corresponds to the beam footprint on the seabed, which is approx. 0.01 $h$ = 0.65 m, where $h$ is the depth of the seabed at the site.

The wreck length measured from the reconstructed 3D model created from the multibeam sonar data was 145 m, the wreck width was 28 m, and the bearing direction from the stern to the bow (bearing angle) was 79.4°. The size of the 3D model matches the real size of the ship very accurately. Precise coordinates of the stern and bow centers in the WGS84 system were also obtained.

Figure 30a shows the reconstructed bathymetry model of S.M.S. Szent Istvan shipwreck seen from its northern side. It is interesting to note how the sides of the shipwreck are very steep, almost vertical. This could be the consequence of a high number of outliers in the point cloud in these areas, due to the fishing nets hanging all over the shipwreck. The ship's propellers, motor shafts and the depression on its aft side are clearly visible in the model, as shown in Figure 30b, which shows the aft of the ship from its southern side with a great similarity to the drawings of the shipwreck in Figure 27a. Moreover, the torn bow part of the ship is shown in Figure 30c, which is also identical to the drawings of Danijel Frka given in Figure 27b.

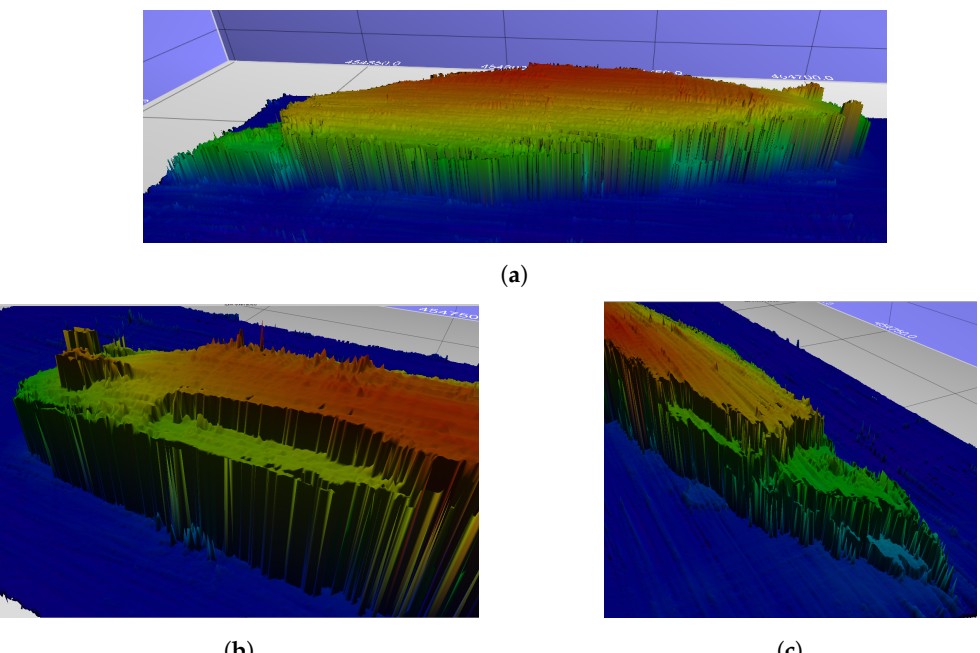

(**a**)

(**b**)    (**c**)

**Figure 30.** (**a**) Bathymetry model of S.M.S. Szent Istvan seen from its northern side. (**b**) Bathymetry model details of the aft seen from its southern side: the depression in ship's hull, propellers, and motor shafts are clearly visible. (**c**) Bathymetry model details of the torn bow seen from its southern side. Based on MBES data collected by ASV PlaDyBath.

Except for having made a 2.5D model of the shipwreck, which can be useful for presentation in museum exhibitions, the resulting point cloud of the model can be used to assess the change of the degradation state of the shipwreck's hull. With future monitoring missions, and using software such as CloudCompare, the changes in shipwreck's hull could be directly and easily tracked and documented.

Another interesting finding and confirmation of the reports we received from the Szent Istvan wreck divers are the opening below the south side of the ship. To record this with the multibeam sonar, its beams had to be tilted in order to catch the morphology of the opening. This made the sonar now able to record the sides of the ship that it could not detect when the sonar beam was not tilted to the side.

Since Qimera, which was used as surface reconstruction software from bathymetric data, reconstructs a 2.5D surface and pairs each $(x, y)$ ordered pair in the horizontal plane to only one height $z$ value, this opening could not be faithfully reconstructed in 3D. Instead, characteristic transverse profiles of the south side of the stern of the ship clearly show an opening 4 m high, extending 100 m along the south side of the ship and entering an average of 3–4 m towards inside the ship, shown in Figure 31a. It is also intersting to note the outliers which were present in some of the ping returns, which are due to the nets laying all over the hull of the shipwreck and even over the above-mentioned opening.

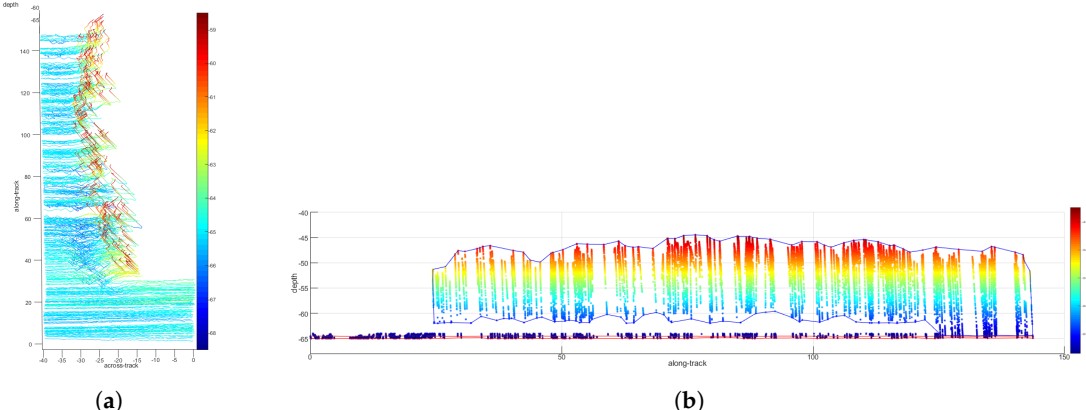

(a)　　　　　　　　　　　　　　　　　　　(b)

**Figure 31.** Acoustic returns of the multibeam sonar: (**a**) 3D line plot of the pings as a spatial representation of the opening below the southern side of the shipwreck. (**b**) Clustered and separated point clouds of the seabed and the shipwreck with the opening clearly visible in between. View is from the southern side of the shipwreck, so aft is on the left.

A proper 3D reconstruction of the shipwreck's side and the opening below was not attainable; a plot of filtered pings in 3D is given in Figure 31a. Depth range is clipped as $z \in [z_{sf}, z_{sf} + 10 \text{ m}]$, where $z_{sf}$ is the depth at which the surrounding seafloor is, and is color-coded. Lower part of the visible shipwreck's hull represents the aft, and the top is the part of the ship closer to the bow, just where the opening ends. Moreover, south is left and north is right. The opening is clearly visible from this plot, as well as the depression left of the aft. This was probably a consequence of the ship hitting the seafloor and thus displacing the sediment.

Further analysis of the ping point cloud shown in Figure 31a consisted of detecting and clustering points belonging to the seafloor and the shipwreck by thresholding depth values. The plot showing these two clustered point clouds is given in Figure 31b. The opening between these two structures is clearly visible as in some previous figures. However, this way, the size of the opening can be assessed numerically, and the degradation level of the shipwreck's metal hull can be numerically represented through time with further monitoring missions.

### 5.5.3. Side-Scan Sonar Survey Results

The shipwreck and the area around it were also recorded by side-scan sonar installed on the LAUV Lupis, again using lawnmower missions along and across the shipwreck. The missions were planned first in relation to the buoy located on the stern of the wreck, and after processing the bathymetric data, the missions were designed in relation to the exact coordinates of the bow and stern. Since the weather conditions on the surface of the sea do not affect the operation of the underwater vehicle, it was used for filming the second day of the expedition on 11 June 2019, when the wind began to rise and create significantly larger waves than on the first day of the expedition.

After the first couple of missions, it was noticed that the navigation accuracy of the AUV Lupis, despite the presence of a DVL sensor (which is usually used to compensate for deviations from the given trajectory due to sea currents), accumulated an error over time. Since the Szent Istvan wreck

contains a huge amount of metal, it caused the compass in the vehicle to deflect, thus negatively affecting the accuracy of the localization. To address this problem, subsequent missions are planned to compensate for vehicle trajectory drift due to external interference (course errors), and based on experience from previous missions.

A total of 7 missions were made, of which 6 were successful:

- M1—Mission along and over the ship. Shooting depth 35 m 6 October 2019
- M2—Mission across and next to the ship. Shooting depth 35 m 6 October 2019.
- M3—Mission north of the ship. Shooting depth 15 m from the bottom 6 November 2019.
- M4—Mission around the ship. Shooting depth 10 m from the bottom 6 November 2019.
- M5—Mission over a ship with sidescan sonar and camera. Shooting depth 40 m 6 November 2019
- M6—Mission over a ship with sidescan sonar and camera. Shooting depth 44 m 6 November 2019

In the case where the AUV Lupis passes directly over the wreck, the part is located in the side-scan sonar blind zone, the so-called propellers, rudders, depression at the rear of the ship, rocking keel, objects on the south side of the stern and bow, six side artillery, nets tangled around the wreck on the bow of the ship, and a broken bow can be clearly seen from the pictures. A composite image of side-scans port and starboard side is given in Figure 32a. The upper left side of the image shows the torn aft; many fishing nets hanging from the ship's side can be seen on the upper right side of the image; the structure of plates along the ships hull can be seen from the left part of the image; keel is visible in the right middle part; propellers and rudders can be clearly seen from the bottom right side of the image.

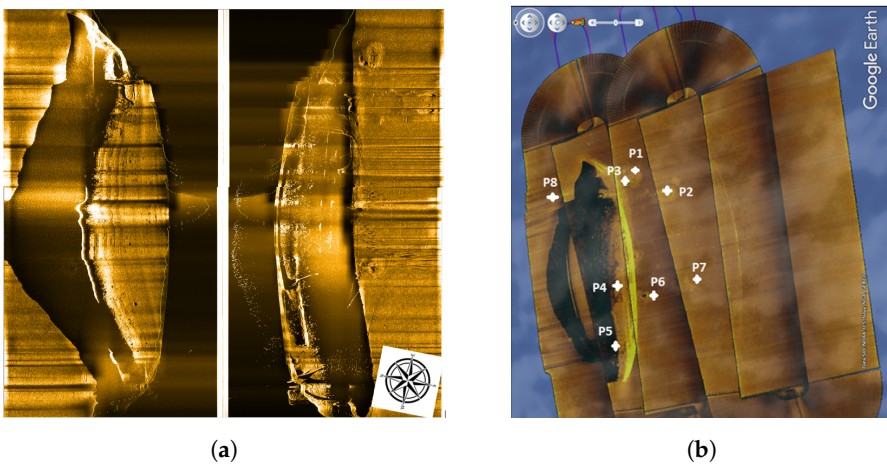

(**a**)                                                                 (**b**)

**Figure 32.** (**a**) Composite of side-scan sonar images of the shipwreck SMS Szent Istvan taken from the mission during which LAUV Lupis crossed directly above the wreck. (**b**) Geolocated mosaic of side-scan sonar data from one of the Lupis AUV missions exported in .kmz format and imported into Google Earth with marked points of interest.

The georeferenced mosaic of side-scan sonar imagery exported to Google Earth is shown in Figure 33b. It is marked with points of interest (POIs) P1–P8. These POIs were detected in all of the side-scan sonar missions, but from different angles. Some of them were known to the underwater archaeologists with experience at this site, but some of them were sent in a report for further analysis with positions relative to the ship and the estimated size of the detected objects. Side-scan sonar images of these objects, as well as details about their position and size, are given in Figures 33–38.

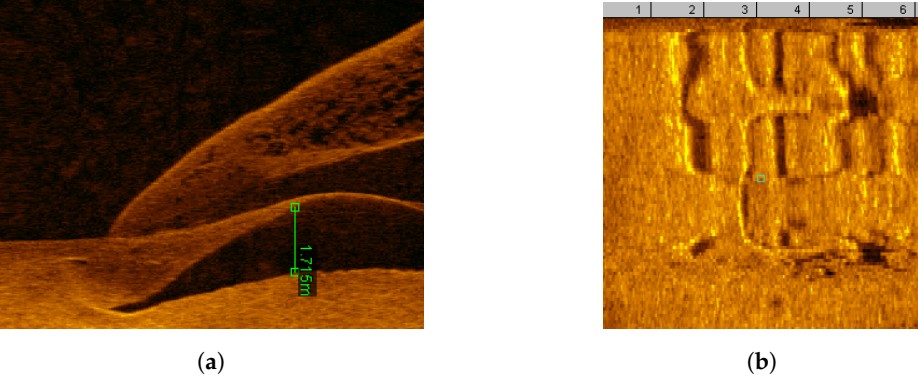

(**a**)    (**b**)

**Figure 33.** (**a**) The bow of the ship and a pile of sand raised by the sinking of the ship (north side). Height 1.7 m. (SI 44 mission). (**b**) Position 1 (P1): Facilities on the south side of the ship. Approx. 25 m from the bow and 10 m south of the ship. Area size 4 m × 6 m.

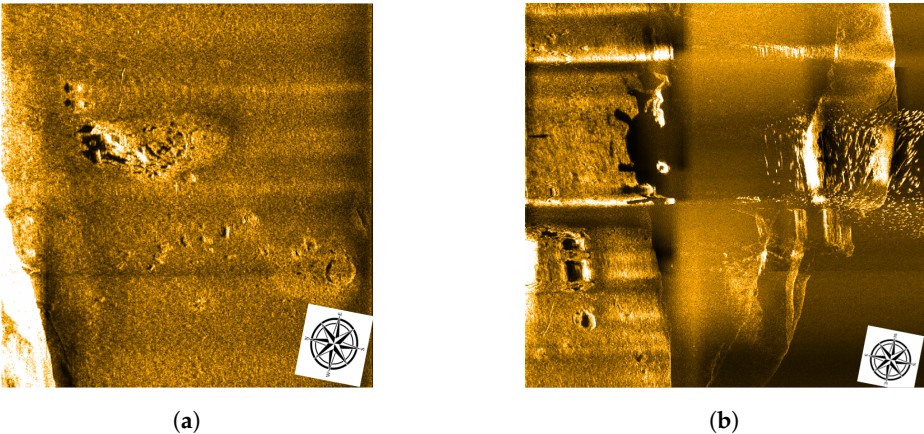

(**a**)    (**b**)

**Figure 34.** (**a**) Position 3 (P3): Interesting objects south of the bow. The upper pile looks like a ship's equipment scattered on the seabed, while the lower pile looks like suitcases measuring 1.3 × 0.5 m. (**b**) Detail of the southern half of the bow of the wreck, nets tangled around the wreck, interesting objects just a few meters south of the top of the bow (left of the wreck in the picture), and the front bow cannons.

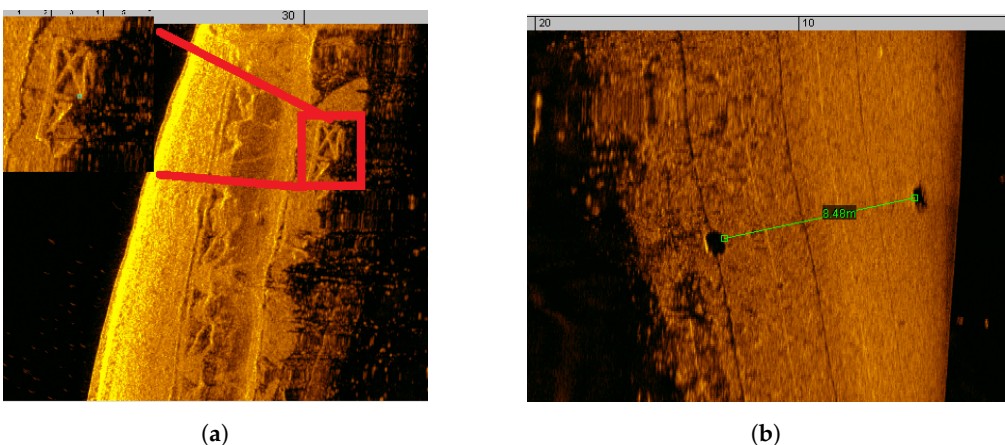

(**a**)    (**b**)

**Figure 35.** (**a**) Position 4 (P4): South side. Two holes (plow)? The size from the sonar image is approx. 3 m each. Position: approx. 70 m from the stern, ie about 85 m from the bow (mission SI 40). (**b**) Position 5 (P5): Water intake holes in the hull symetricaly positioned on the port and starboard side of the hull. Diameter 1 to 1.5 m. Position: upper side of the formwork, approx. 40–42 m from the last point of the stern of the ship. View of the north hole, another symmetrical hole on the south side, just before the start of the rocking keel.

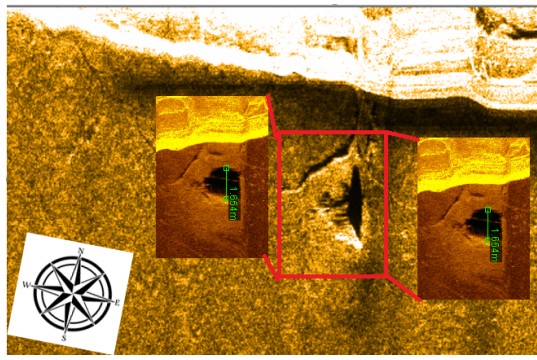

**Figure 36.** Position 6 (P6): Details of an interesting object (which casts an acoustic shadow) 3 m long and 1.5 m high on the south side of the stern taken in two missions.

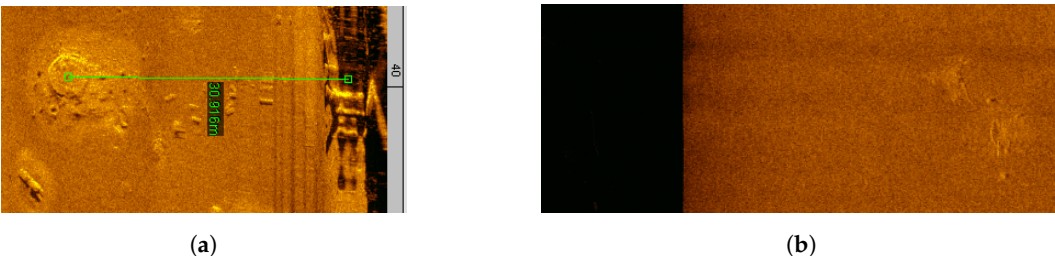

| (**a**) | (**b**) |

**Figure 37.** (**a**) Position 2 (P2): Same group of objects 1: approx. 40 m from the bow and extending from the ship to approx. 30–35 m south of the ship. (**b**) Position 7 (P7): Object about 40 m from the ship on the south side about 60 m from the stern, i.e., 90 m from the bow. These could perhaps be the remains of old fishing nets.

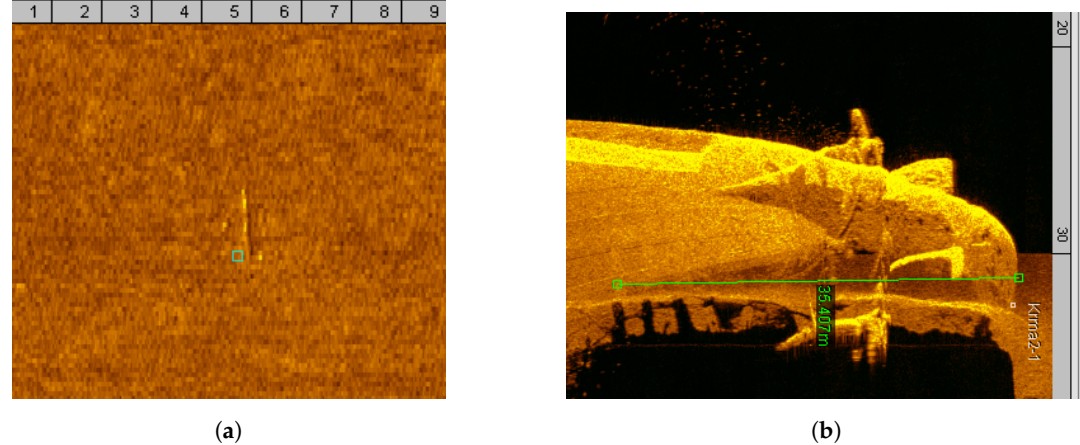

| (**a**) | (**b**) |

**Figure 38.** (**a**) Position 8 (P8): North side of the ship, mission 75 m range, one bar-shaped object about 2.5 m long: position: about 20 m from the ship and about 40 m from the bow. (**b**) Aft part. Propeller, rudders and cannons. Stern cannons are positioned at 29 m, 32, and 35 m from the stern.

### 5.5.4. AUV's Visual Inspection Results

In addition to side-scan sonar, the missions that used AUV Lupis to record the wreck of SMS Szent Istvan and surroundings used an integrated CCTV 1.4 MP camera for visual inspection of the wreck in missions where the vehicle is planned to move a few meters above the wreck. The mission of the AUV Lupis was planned to move at a constant depth and along the ship's line from the stern to the bow, which we determined precisely from the bathymetric 3D model. Below are a couple of examples of interesting things captured by AUV's camera. Since the AUV had an altitude of less than 10 m when crossing over the wreck, its LED flash burned the middle of each frame quite a bit. Red and yellow hues were suppressed as expected, and the images are bluish, which was expected because of the depth at which the shipwreck lays.

An example of an image of the shipwreck's hull taken by AUV Lupis camera is shown in Figure 39. The original image is shown in Figure 39a. Figure 39b shows the original burnt image filtered by the contrast-limited adaptive histogram equalization (CLAHE) [32] in LAB color space. Figure 39c shows the result of applying the adaptive histogram equalization (AHE) algorithm [37] to YCrCb color space of the original image. Lastly, Figure 39d shows the result of applying the adaptive histogram equalization (AHE) algorithm [37] to YUV color space of the original image. It is noticeable that CLAHE, as an adaptive algorithm for cropping and equalizing the histogram of the brightness of the image, gives the best results with the fewest artifacts, either by brightness or color change.

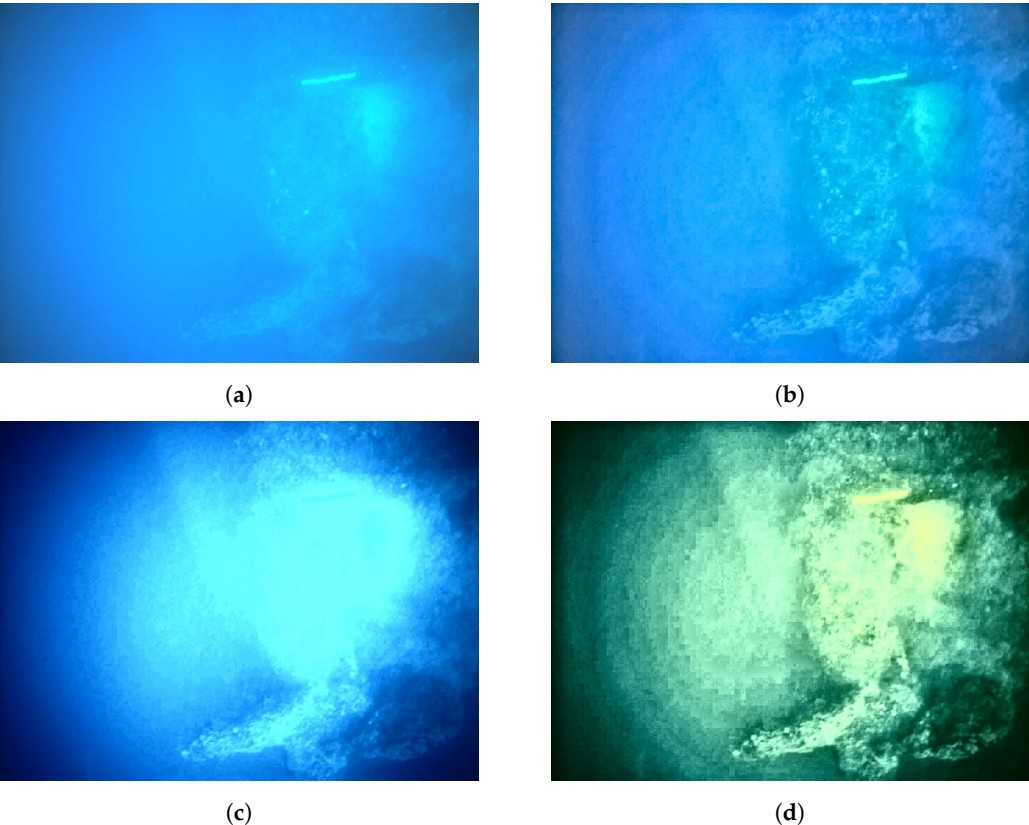

**Figure 39.** An example of brightness equalizing algorithms applied to visual data recorded by AUV Lupis. (**a**) Original image. (**b**) Image filtered by CLAHE algorithm in LAB color space. (**c**) Image filtered by HEQ algorithm in YCrCb color space. (**d**) mage filtered by HEQ algorithm in YUV color space.

Moreover, video frames captured by the AUV Lupis camera were used to generate 3D models and orthophoto projections (see Figure 40) in Agisoft Metashape software with high settings on photo alignment, point cloud, and mesh. Area covered by this orthophoto is $4 \times 80$ m. Although the 3D model is not georeferenced (though with additional processing of AUV Lupis logs, each frame can be paired with the estimated AUV Lupis position), nor does it represent the full picture of the bottom of the sunken ship Szent Istvan, it provides insight into how fast and easy autonomous underwater robots can be used. In a short period of time, they record the area that divers would record for hours or days, due to the limited duration of the dive at the depth of almost 70 m at which the shipwreck lays.

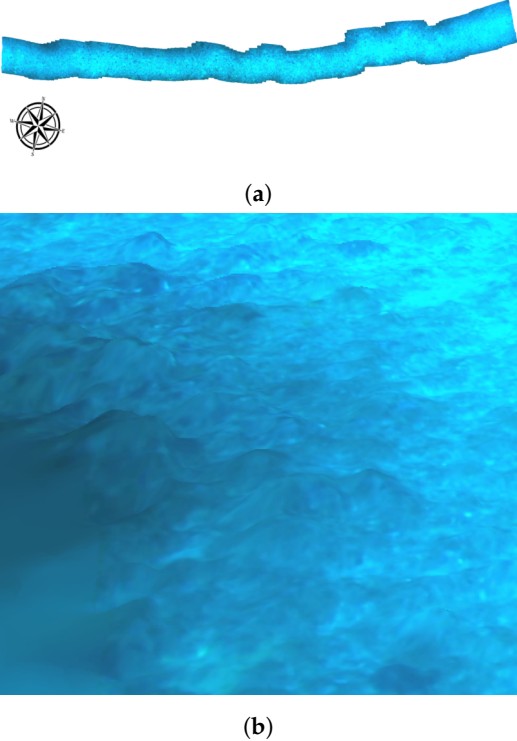

(**a**)

(**b**)

**Figure 40.** (**a**) Orthofoto mosaic of the top of the wreck obtained from the AUV Lupis camera images. Area covered: 4 × 80 m. (**b**) A detail from the orthophoto showing level of detail of the orthophoto and biofouling on the Szent Istvan shipwreck hull.

## 6. Discussion

One of the objectives of the BLUEMED project was to offer novel solutions for preservation of the degrading ancient UCH sites but in the digital realm. This enables permanent availability and presence of the documented sites in a form of 3D models. Moreover, these 3D models set in a virtual reality were planned to be used for the so-called dry dive visits to the sites. This entails emulating the dive to the UCH site by means of the virtual reality (VR) set. The VR device introduces two separate implementations from the user's point of view: in the first example, users will conduct a semi-immersive visualization using a full HD display based on passive 3D technology, as seen in Figure 41a. Users will engage with the device using a multi-touch screen tablet with a user interface that offers all the input features required for navigating the 3D world and accessing multimedia details. The visualization may also be carried out in an interactive setting by HMD (head-mounted display) technology, as seen in Figure 41b. In addition, an HTC Vive with wireless handheld controls was used for this reason for interaction.

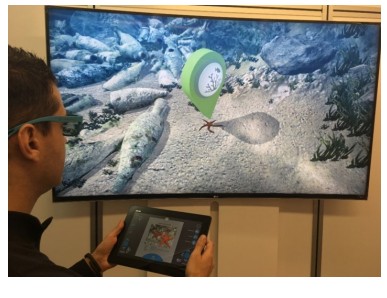 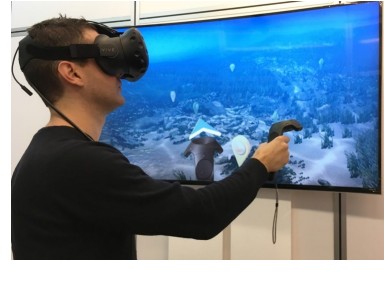

(**a**)                    (**b**)

**Figure 41.** (**a**) VR semi-immersive visualization. (**b**) VR immersive visualization. Courtesy of Fabio Bruno.

Augmented reality diving experience is another innovative thing which was the result of the BLUEMED project. It merged the georeferencing of the UCH site with acoustic underwater localization of the divers. Collocation of the photogrammetric model with the georeferenced bathymetric 2.5D manifold yielded georeferenced coordinated of the UCH sites of interest. Moreover, a short baseline (SBL) system, mounted onto LABUST's modified ASV, was used for acoustic localization of the divers. The coordinate frame transforms from GPS-enabled ASV to the diver's estimated relative position w.r.t. The ASV and the diver could thus be globally localized. This meant that the diver's position was known w.r.t. the known position of the UCH site. The schematic of the above-mentioned approach is shown in Figure 42.

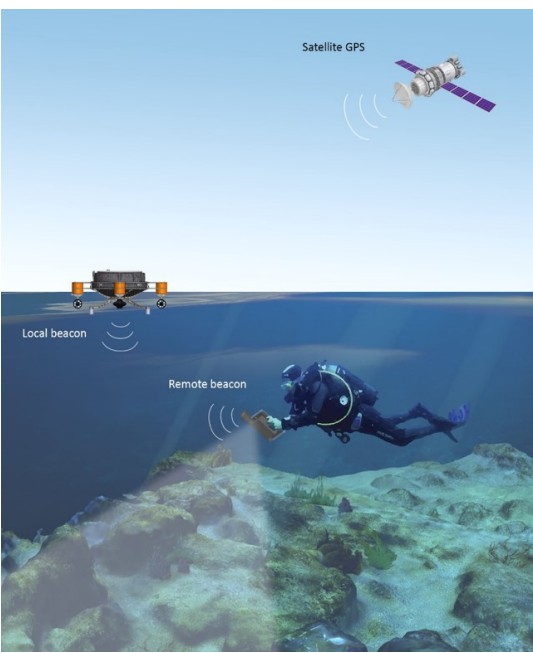

**Figure 42.** Underwater localization of diver(s) diving at the underwater archeological site. Courtesy of Fabio Bruno.

An underwater tablet with mounted acoustic communication device was designed by our colleagues from University of Calabria (UNICAL), led by Fabio Bruno. LABUST developed an ASV designed to carry the SBL system developed by UNICAL. An example of this tablet's graphical user interface (GUI) is shown in Figure 43a. The diver is presented with the data about the dive duration, depth, temperature, and heading. However, all this information is available on commercial diver watches. The added value of such an underwater tablet application was to offer the diver the possibility to know his/hers global position, have access to the list of the POIs along the site, but also gain notification-like suggestions about interesting site's details nearby, as well as information in textual or audio form. Detailed tests of the underwater localization and the augmented reality (AR) system were performed in the Summer of 2018 in Cavtat (Croatia), as shown in Figure 43b.

Another important aspect which is emphasized here is the advantage of using marine robots in UCH site survey tasks. The initial investment needed to purchase such systems is much higher than the case of the immediate cost of hiring divers. However, in the long run, the autonomous systems are much cheaper. They can operate much longer than human divers in various environmental conditions (cold and/or water, great depths, contaminated areas etc.). Regarding UCH sites, their deployment is especially significant and advantageous at the sites that are at depth of more than 50 m as is e.g., Szent Istvan shipwreck. Since we worked at this site alongside professional technical divers, we can directly compare the efficiency of both methodologies. It took the ASV and the AUV around 20 min to survey the area of $200 \times 75$ m and thereby gather millions of data points by MBES, SSS and AUV's camera. To reach this level of precision, any manual work would surely last a few orders of

magnitude longer. Furthermore, the divers worked in 90 min shifts. Out of those 90 min, they were effectively working only 30 min, while the remaining time was spent on the descent and ascent. Even if a diver could gather the same amount of photos along the shipwreck in 30 min as the AUV did, this still means that the AUV is 4–5 times more efficient. Moreover, the use of an ROV for taking oblique photos of the whole wreck seems now like a preferable option if the whole Szent Istvan shipreck is to be reconstructed as a 3D photogrammetric model.

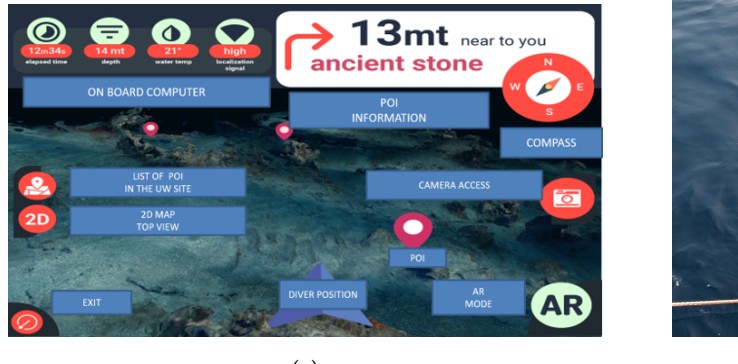
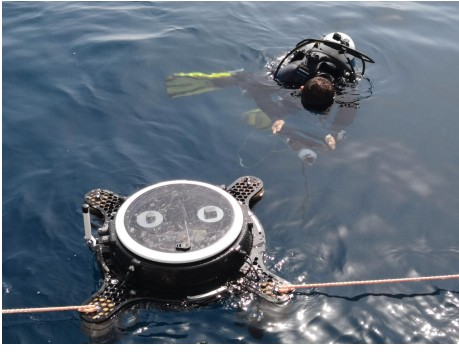

| (**a**) | (**b**) |

**Figure 43.** (**a**) Augmented reality graphical user interface. Courtesy of Fabio Bruno. (**b**) A diver testing the AR system with the underwater tablet and the ASV developed for underwater localization aiding at the Cavtat (Croatia) UCH site.

## 7. Conclusions

A great potential for use of the autonomous marine vehicles in remote sensing surveys is presented in this article. Here, three marine vehicles' and one unmanned aerial vehicle's use in nine underwater archaeology related autonomous remote sensing survey missions is described through the surveying methodologies which yielded the presented results. The efficiency of using autonomous vehicles for such applications, data quality, quantity in unit time, as well as the precision of georeferencing the data, especially on larger areas as in the authors' case studies, has been thus shown to far surpass the divers' performance.

The precision of bathymetry georeferencing ranged from 1 cm when the local base station corrections were available to 10 cm when only Applanix INS with Trimble GPS antennae were used. The resolution of bathymetric models ranges from 20 cm for shallow areas, where most of the UCH sites were located at up to 50 cm for deeper parts. Position estimation of the AUV and SSS imagery were around 1 m after smoothing the position estimation in the post-processing phase staring from the end to the beginning of the dive. The AUV has been shown to be at least 4–5 times more efficient for visual inspection of the wrecks at depths over 50 m compared to technical divers. Photogrammetric 3D models generated from photos taken by AUV, ROV and UAV show that even 20 min autonomous survey missions can yield high-resolution models for various applications.

It is important to note that this was the first time that any of the chosen sites were documented by sonar technologies or autonomous marine vehicles. The main objective of the these surveys was to document and assess the current state of the sites and to establish a foundation which future monitoring operations could be built upon and compared with. Moreover, going beyond mere documentation and physical preservation, examples of using these results for digital preservation of the sites in augmented and virtual reality are also presented.

Looking forward, the research goal of the authors is to develop higher levels of vehicles' autonomy, which would encompass autonomous exploration/inspection path (re)planning based on the online processed sensor data and artificial intelligence, in order to optimize the performance of the autonomous vehicles even further.

**Author Contributions:** Conceptualization, N.K., A.V., and N.M.; Formal analysis, N.K., A.V., Đ.N. and K.Z.; Methodology, N.K., A.V., Đ.N. and K.Z.; Software, N.K., and Đ.N.; Supervision, A.V., N.M., K.Z.; Validation, N.K., A.V., Đ.N. and K.Z.; Writing—original draft, N.K., A.V., N.M. and K.Z.; All authors have read and agreed to the published version of the manuscript.

**Funding:** This research is sponsored by Croatian Science Foundation Multi Year Project under G.A. No. IP-2016-06-2082 named CroMarX; the Foundation of the Croatian Academy of Science and Arts and the EU Regional Development funded project DATACROSS under G.A. No. KK.01.1.1.01.0009; the H2020-INFRAIA funded EUMarineRobots project under G.A. No. 731103; and the European Regional Development Fund co-financed HEKTOR project KK.01.1.1.04.

**Acknowledgments:** The authors would also like to thank Milan Marković, the former mechanical engineer of LABUST, for the design and construction of the ASV PlaDyBath, as well as for all the help during the recordings of all the underwater archaeological sites in the scope of BLUEMED project; Fabio Bruno and his team from the University of Calabria for the close collaboration during BLUEMED project's UCH sites data collection as well as merging of the bathymetric 2.5D manifold with their 3D photogrammetry model; and Nikica Kokir for all the technical and logistics help during data collection at the Plitvice national park.

**Conflicts of Interest:** The authors declare no conflict of interest.

## Abbreviations

The following abbreviations are used in this manuscript:

| | |
|---|---|
| 2.5D | 2.5-dimensional |
| AHE | Adaptive histogram equalization algorithm |
| AR | Augmented reality |
| ASV | Autonomous surface vehicle |
| AUV | Autonomous underwater vehicle |
| CLAHE | Contrast-limited adaptive histogram equalization algorithm |
| DSLR | Digital single-lens reflex camera |
| DVL | Doppler velocity logger |
| ESC | Electronic speed control |
| FXTI | Fathom-X tether interface |
| GPS | Global Positioning System |
| GSM | Global System for Mobile Communications |
| GUI | Graphical user interface |
| HF | High frequency |
| IMU | Inertial measurement unit |
| LABUST | Laboratory for Underwater Systems and Technologies |
| LAUV | Lightweight autonomous underwater vehicle |
| LF | Low frequency |
| LSTS | Laboratório de Sistemas e Tecnologia Subaquática |
| MBES | Multibeam echosounder |
| PlaDyBath | Platform for dynamic bathymetry |
| PlaDyPos | Platform for dynamic positioning |
| POI | Point of interest |
| ROS | Robot operating system |
| ROV | Remotely operated vehicle |
| SBL | Short baseline |
| SLAM | Simultaneous localization and mapping |
| SSS | Side-scan sonar |
| UAV | Unmanned aerial vehicle |
| UCH | Underwater cultural heritage |
| UNICAL | University of Calabria |
| UNIZG-FER | University of Zagreb Faculty of Electrical Engineering and Computing |
| USBL | Ultra-short baseline |
| VR | Virtual reality |

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
