# Peer review of "Marine Robots Mapping the Present and the Past: Unraveling the Secrets of the Deep"

_remotesensing, doi:10.3390/rs12233902_

Round 1

Reviewer 1 Report

The documentation is really extensive and interesting.

 I waited for a metrical approach in the experiments and an evaluation of the accuracy of the results either in terms of instruments and techniques or in terms of 3d models accuracy.

After all in the abstract and the introduction the authors talked about new approach and integration between marine robots and remote sensing technologies for "more accurate, efficient and precisely georeferenced " results.

Effectively the authors shown good quality results in terms of 3d models visualization but there is not mention about the precision of procedures and the accuracy of results.

I strongly suggest a major revision of the paper where the accuracy of the results and of the 3D models should be demonstrated. In this case the value of site documentation will be proved even for technical usage of the mapping.  

Moreover it should be interesting even a comparison between marine robots and human divers in terms of potentiality and accuracy in final results.

Reviewer 2 Report

The work contains many examples of underwater surveys carried out with different instrumentation and methodologies. The figures produced are very significant and demonstrate the authors' experience in the design and execution of underwater surveys, but also the lack of application of the processing and filtering techniques of the acoustic data coming from the Side Scan Sonar.

Indeed the morpho-acoustic data certainly are high resolution ones but they need a more intense post-processing phase  to improve the quality of the outputs. For instance, in the case of Side Scan Sonar mosaics, it would be useful to apply a botto-trackin correction and some filter as the TVG.

However the paper is described in a way that looks more like an advertising brochure rather than an article supposed to be in a scientific journal.

Reviewer 3 Report

The topic of the article is interesting and relevant to research communities within both remote sensing and archaeology. However, the manuscript in its current form seems more like a list of activities with mini-reports, than a scientific paper. It lacks a research question or aims/goals section that gives the work a clear direction or purpose. I recommend that the authors include that in a revised version, and additionally a section in the discussion/conclusion that states the scientific contributions of the work/article towards the research question(s), aims/goals. 

Comments to the text referring to line numbers:

50: "pre-disturbed". Do you mean non-intrusive? The site may well be disturbed before archaeological surveying. Fishing, divers etc.

53: A bit unclear what is meant by "direct contact with human divers".

62-70: What is the argument here? Underwater positioning capabilities have improved, enabling better precision - yes. But the paragraph is a bit confusing and requires rephrasing. Perhaps you could state that AUVs can maintain acceptable positioning precision by combining GPS fix with DVL-lock and IMU. USBL aided navigation, requiring a surface vessel or buoy with GPS-positioning, may further increase precision in navigation even in deeper waters. Not sure that I understood what is being communicated in this paragraph.

89: Were the sites not recorded before at all, or was this the first time they were recorded using AUVs and sonar? I suspect you mean the first, but it can be a bit unclear.

91: A research article would normally have a research question here. As it stands now the article appears to be a presentation/review of several projects that LABUST/the authors have conducted. If they are case studies, a problem or question they are meant to elucidate must be presented.

98: The history of LABUST is interesting and impressive. However, the direct relevance of this section to the topic of the article is not apparent. It would be better if this section presented the state of the art, referring more extensively to relevant research in the field of Marine Robots mapping UCH. 

169: What is meant by previously? Is it real-time or near real-time so the operator can take actions in case quality or coverage is poor?

191: 10mm resolution at what range?

193: Perhaps a short description of the measurements and output from multibeam sonar? Bathymetry (point cloud) + backscatter.

199-222: A bit too elaborate technical description of LAUV. Could be shortened with a reference to a technical data sheet or producer's webpage. Reads a bit like a catalogue.

236: Output? Georeferenced sonar images. 

238: superior to what?

246-247: This type of claim should either be avoided or substantiated by convincing arguments. It doesn't seem very scientific.

262: The design of preceding variants does not seem very relevant.

271: A remote control unit with a built-in screen, I assume.

272: Updated from what? Is that relevant?

279-296: If results of any of these collaborations have been published, they should be referenced for the reader to find them. 

317: In figures 7 and 8 there is reference to MBES data. If the DVL was used to acquire bathymetric data, this should be corrected. Also, resolution/swath of DVL-derived bathymetry should be described.

322: A more detailed account of the results should be provided for the reader, not just figures. Did the data contribute to the archaeological end-user's needs and wants? How was it helpful for them to use this technology? Since there is no Discussion section for each case, a few sentences could be included in the respective results section to help the reader understand the relevance of these technologies for marine archaeology. The exception is the Szent Istvan shipwreck (4.5) which is well described, demonstrating how different sensors and acquisition strategies were used strategically and adaptively to answer knowledge needs. Also problems, errors or shortcomings were described in 4.5, helping the reader to better assess the methods and how appropriate they are for the initial purposes. This was generally lacking in the other cases. It would heighten the value of the article to mention challenges and problemativ issues as well. 

Figure 8: DVL-derived bathymetry or MBES?

Figure 37: This figure does not illustrate well the quality and LOD of the Orthophoto mosaic. Perhaps an insert of a selected area should be included in the figure?

Round 2

Reviewer 1 Report

The improvement of the paper is sufficient to justify the publication.

It does not represent a really high accuracy surveying and mapping but it is appropriate for the purposes thought by the authors.

Reviewer 2 Report

The new version of the paper meets the correct criteria of a scientific journal

Reviewer 3 Report

Dear editor and authors,

The authors have responded to my suggestions and comments to the first version, and have changed/improved the article to my satisfaction. I still think a better presentation of state of the art within marine robotics applied to UCH would have provided a wider context for the work presented, and underscored the importance of the long term research goals of the authors. However, this is not a critical issue in my opinion. I believe the revised version can be accepted in its present form, and that the article will be a valuable contribution to an important field of research.